# A holistic platform for accelerating sorbent-based carbon capture

Charithea Charalambous[1,8], Elias Moubarak[2,8], Johannes Schilling[3,8], Eva Sanchez Fernandez[4], Jin-Yu Wang[1], Laura Herraiz[1], Fergus Mcilwaine[1], Shing Bo Peh[1], Matthew Garvin[1], Kevin Maik Jablonka[2], Seyed Mohamad Moosavi[2], Joren Van Herck[2], Aysu Yurdusen Ozturk[5], Alireza Pourghaderi[6,7], Ah-Young Song[6,7], Georges Mouchaham[5], Christian Serre[5], Jeffrey A. Reimer[6,7], André Bardow[3], Berend Smit[2✉] & Susana Garcia[1✉]

Reducing carbon dioxide ($CO_2$) emissions urgently requires the large-scale deployment of carbon-capture technologies. These technologies must separate $CO_2$ from various sources and deliver it to different sinks[1,2]. The quest for optimal solutions for specific source–sink pairs is a complex, multi-objective challenge involving multiple stakeholders and depends on social, economic and regional contexts. Currently, research follows a sequential approach: chemists focus on materials design[3] and engineers on optimizing processes[4,5], which are then operated at a scale that impacts the economy and the environment. Assessing these impacts, such as the greenhouse gas emissions over the plant's lifetime, is typically one of the final steps[6]. Here we introduce the PrISMa (Process-Informed design of tailor-made Sorbent Materials) platform, which integrates materials, process design, techno-economics and life-cycle assessment. We compare more than 60 case studies capturing $CO_2$ from various sources in 5 global regions using different technologies. The platform simultaneously informs various stakeholders about the cost-effectiveness of technologies, process configurations and locations, reveals the molecular characteristics of the top-performing sorbents, and provides insights on environmental impacts, co-benefits and trade-offs. By uniting stakeholders at an early research stage, PrISMa accelerates carbon-capture technology development during this critical period as we aim for a net-zero world.

Solid adsorbent-based carbon capture can leverage modern reticular chemistry to synthesize millions of possible adsorbents[7], including around 100,000 metal–organic frameworks (MOFs)[8,9]. To fully explore this potential, we must move beyond the conventional sequential, time-consuming trial-and-error approach. Computational groups have initiated material genomics to accelerate discovery, generating materials in silico and predicting their adsorption properties through molecular simulations[10–13]. Although these predictions are promising, their impact has been limited because they often assume that a few basic adsorption properties (for example, Henry selectivity or carbon dioxide ($CO_2$) capacity) suffice to evaluate material's performance. The optimal material depends on specific process requirements, scale-up, location and life-cycle assessment (LCA)[4,5,14–20]. The lack of system-level contextualization has hindered stakeholder engagement in materials discovery. A holistic approach is needed to link material properties with process design and techno-economic analysis (TEA). An LCA further evaluates environmental impacts beyond climate change, ensuring that the carbon-capture plant's construction and operation do not result in higher $CO_2$-equivalent ($CO_2$e) emissions than it mitigates over its lifetime[21].

## The PrISMa platform for carbon capture

The PrISMa (Process-Informed design of tailor-made Sorbent Materials) platform (Fig. 1) allows for the interrogation and screening of materials for a given case study, which is defined by the $CO_2$ source, the destination of the $CO_2$ (sink), the capture technology, the available utilities and the geographical region (Extended Data Table 1). In the materials layer, we use experimental data or crystal structures to predict the adsorption thermodynamics of flue gas components ($CO_2$, nitrogen ($N_2$) and water ($H_2O$)) through molecular simulations. These thermodynamic data and process and equipment data serve as input for the process layer, where we compute parameters such as purity, recovery, productivity and energy requirements. In the TEA layer, we assess the economic and technical viability. The LCA layer then evaluates the environmental impacts over the plant's lifetime.

[1]The Research Centre for Carbon Solutions (RCCS), School of Engineering and Physical Sciences, Heriot-Watt University, Edinburgh, UK. [2]Laboratory of Molecular Simulation (LSMO), Institut des Sciences et Ingénierie Chimiques, École Polytechnique Fédérale de Lausanne (EPFL), Sion, Switzerland. [3]Laboratory of Energy and Process Systems Engineering (EPSE), ETH Zurich, Zurich, Switzerland. [4]Solverlo Ltd, Dunbar, UK. [5]Institut des Matériaux Poreux de Paris, Ecole Normale Supérieure, ESPCI Paris, CNRS, PSL University, Paris, France. [6]Materials Science Division, Lawrence Berkeley National Laboratory, Berkeley, CA, USA. [7]Department of Chemical and Biomolecular Engineering, University of California Berkeley, Berkeley, CA, USA. [8]These authors contributed equally: Charithea Charalambous, Elias Moubarak, Johannes Schilling. ✉e-mail: berend.smit@epfl.ch; s.garcia@hw.ac.uk

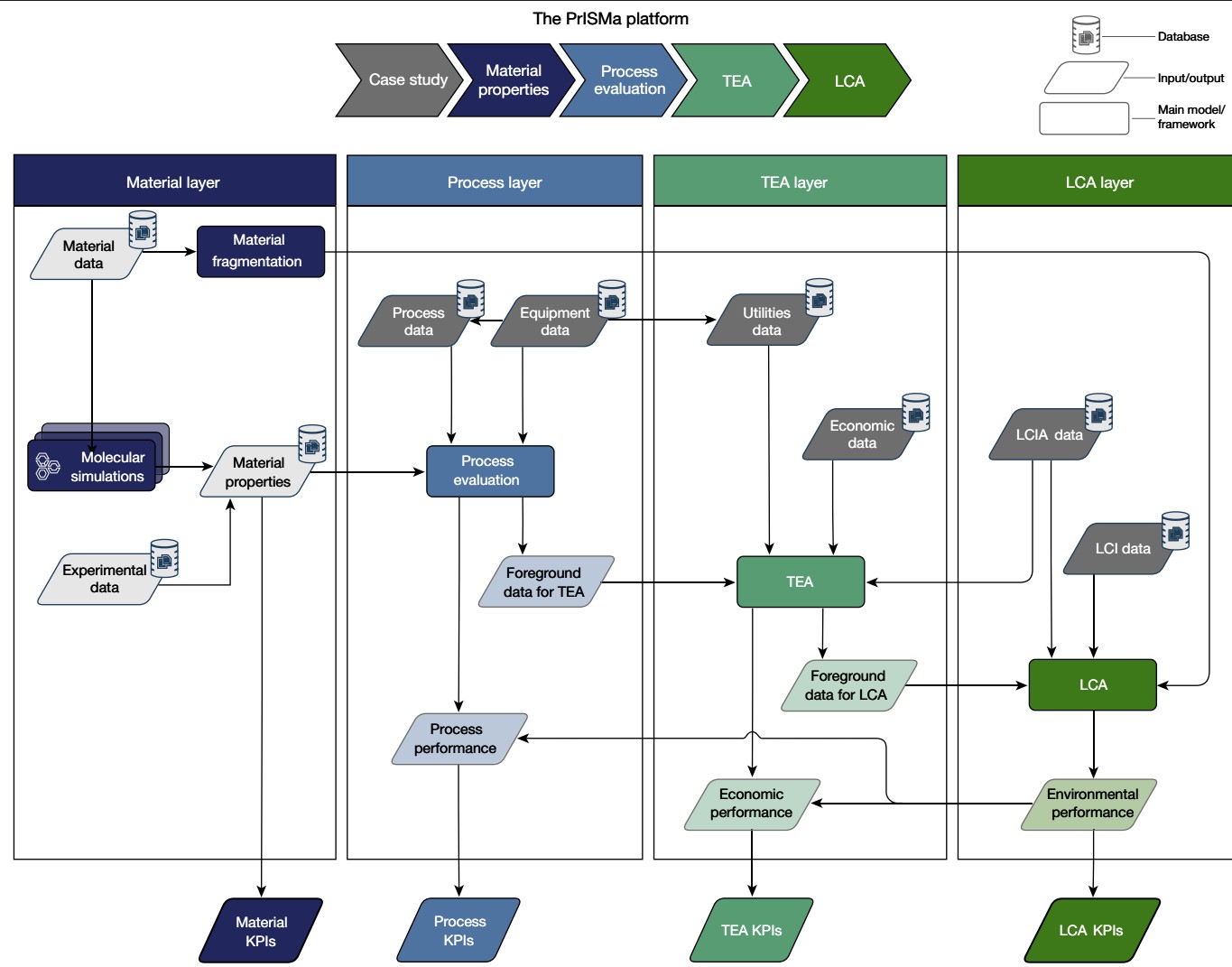

**Fig. 1 | The PrISMa platform screens solid sorbents for $CO_2$-capture applications.** The data flowchart of the four layers of the platform. The figure illustrates the links between the LCA, the TEA, process evaluation and material characterization. On the basis of the crystal structure of a sorbent material, we evaluate its performance for a specific carbon-capture process, connecting a $CO_2$ source with a $CO_2$ sink in a region of the world, using a total of 50 KPIs. The platform integrates databases on material properties, process design parameters, utilities data, economic data, life-cycle impact assessment (LCIA) data and life-cycle inventory (LCI) data. A detailed description of the methods used in each layer can be found in Supplementary Information Section 3. Our interactive visualization tool provides access to all KPIs for all case studies and more than 1,200 materials[25].

Following this holistic approach, the platform identifies top-performing materials for further study. These materials can then undergo more detailed process modelling and investigation (for example, sorbent durability and manufacturing) to advance the technology to pilot and demonstration scales.

## Informing stakeholders' perspectives

The PrISMa platform's modular structure allows us to consider various stakeholders' perspectives. For any combination of source, sink, technology, utilities and region, we compute a list of 50 key performance indicators (KPIs; Supplementary Table 5). A Spearman analysis (Extended Data Fig. 1) helped us identify six reference KPIs that capture the most important trends (Extended Data Table 2).

Let us first focus on one case study: capturing $CO_2$ using a temperature vacuum swing adsorption (TVSA) process (with a vacuum pressure of 0.6 bar) from a cement plant located in the UK. The captured $CO_2$ is compressed and sent for geological storage. In Fig. 2, we compare the performance of the materials with the monoethanolamine (MEA) benchmark[22] (Supplementary Information Section 4); many materials outperform the benchmark for the different process, TEA and LCA KPIs.

The net carbon avoidance cost (nCAC) is the KPI that quantifies the cost of avoiding $CO_2$ emissions into the atmosphere over the plant's life cycle. The nCAC is not the only criterion, and evaluating materials across all KPIs and from all stakeholders' perspectives is important. Figure 3 highlights the top-performing materials for a given KPI and their ranking on the other KPIs across the platform. The comparison of the material rankings in Fig. 3 illustrates the complexity of selecting an optimal material; the top ten for a given KPI do not necessarily perform well for the other KPIs.

From an engineering perspective, we are interested in identifying the best technology. Figure 4a compares the nCAC of the 20 top-performing materials for the 3 process configurations and 3 $CO_2$ sources. For all three technologies, we find materials that outperform the benchmark for coal and cement. For cases with a low $CO_2$ concentration in the feed stream (for example, natural gas combined cycle (NGCC) power plants), the vacuum step in the process configuration reduces the cost, but no materials are identified with a lower nCAC than the MEA benchmark.

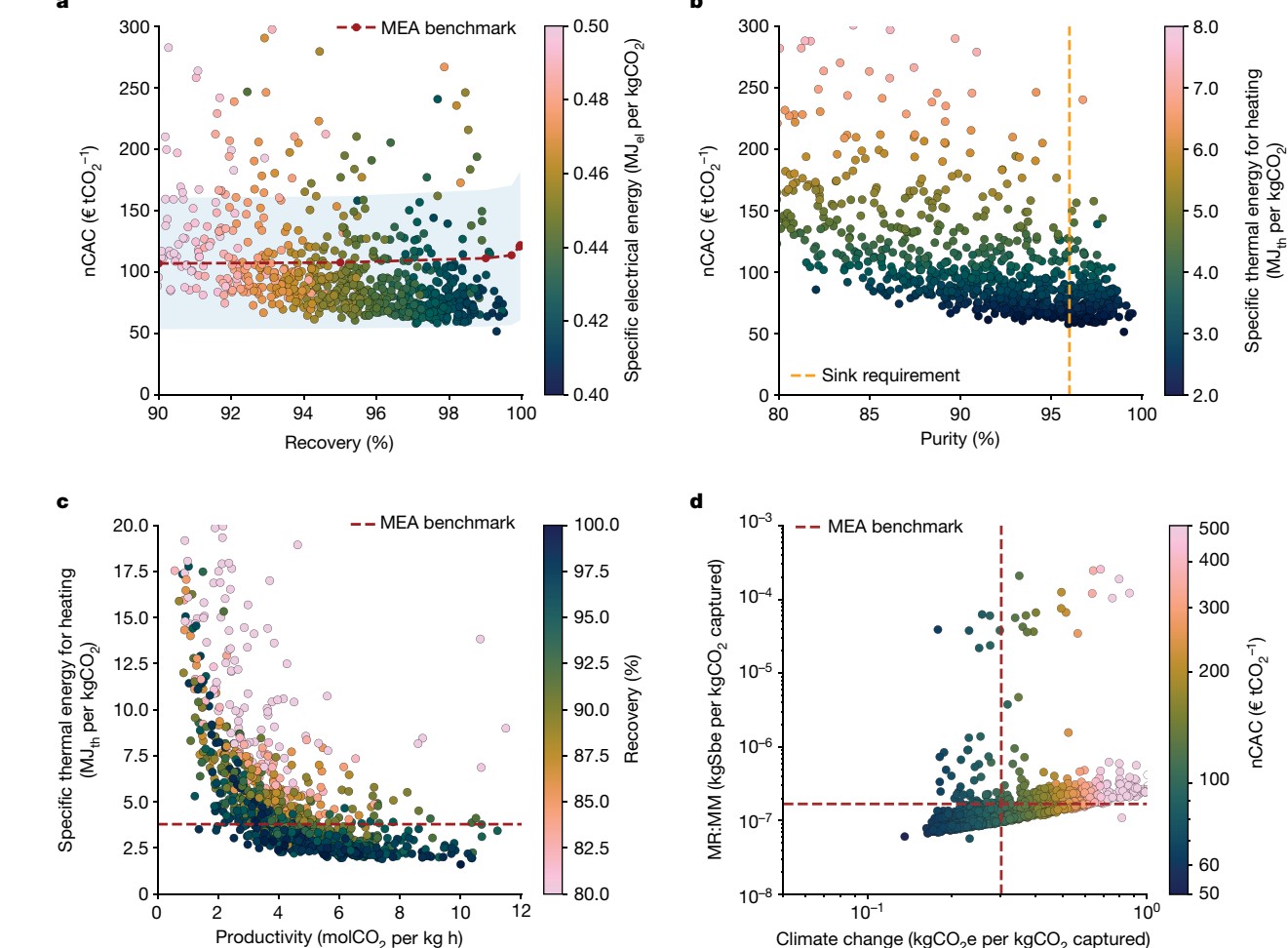

**Fig. 2 | Materials performance for a TVSA carbon-capture process at 0.6 bar added to a cement plant in the UK. a,** The nCAC versus recovery, with colour coding the specific electrical energy consumption. **b,** The nCAC versus purity, with colour coding the specific thermal energy consumption. **c,** Specific thermal energy consumption for heating versus productivity, with colour coding the recovery. **d,** MR:MM versus climate change, with colour coding the

nCAC. Sbe, antimony equivalent. Our visualization tool[25] gives an interactive version of this graph. The dotted lines in **a**, **c** and **d** show the MEA benchmark (Supplementary Information Section 4). In **b**, the vertical orange dotted line gives the purity required for geological storage (>96%) and in **a**, the blue shaded area gives the uncertainty. Each dot represents the corresponding KPI of a material.

The vacuum step increases the purity of the product stream. This increase is achieved by rapidly purging the weakly adsorbed components from the column's gas phase after the adsorption step but at the expense of a lower recovery than a temperature swing adsorption (TSA) process. Figure 4b shows that with the vacuum step, most materials exceed 96% purity, whereas for TSA, only a few materials meet this requirement for geological storage. Therefore, we focus on operating TVSA with 0.6 bar for the cement and coal and the TVSA with 0.2 bar for the NGCC.

After optimization, many more materials meet the purity requirement (Supplementary Information Section 10.3.3). Optimization lowers the nCAC by about €7 tCO$_2^{-1}$ (about 12%) for a TVSA process (cement in the UK) and about €9 tCO$_2^{-1}$ (about 14%) for a TSA process and reduces the differences between the various process configurations. Importantly, we see that the ranking of the top-performing materials has not been impacted significantly.

Running a carbon-capture plant inherently produces emissions of CO$_2$ and other greenhouse gases owing to an increased demand for energy and materials. The environmental manager's perspective focuses on maximizing the captured CO$_2$ while simultaneously minimizing these associated CO$_2$e emissions and other possible environmental impacts.

The effective recovery (Fig. 4d) adjusts the process recovery for the CO$_2$e emissions associated with building and operating the carbon-capture plant, using the climate change KPI (Fig. 4c). For some materials, we find that the climate change KPI is >1 kgCO$_2$e per kgCO$_2$ captured (Extended Data Fig. 2a). This indicates that the capture process with these materials emits more CO$_2$e over the plant's lifetime than the total amount of CO$_2$ captured. Several factors can contribute to this result. For example, some materials have a very low CO$_2$ working capacity, resulting in high material and energy demands. Some others, with relatively good working capacities and moderate heat demands, contain metals such as gold or rhodium. The climate change impact of synthesizing such materials is so significant that it leads to a climate change KPI >1 kgCO$_2$e per kgCO$_2$ captured. An important environmental KPI is the material resources:metals/minerals (MR:MM), which relates to the use of minerals and metals resources. In Extended Data Fig. 3, we compare the ranking of materials based on their constituent metals, focusing on some abundant metals (magnesium, zinc and manganese) and rare metals (copper, lutetium and silver). The MR:MM ranking will be poorer if a greater amount of the corresponding MOF is required to remove a unit of CO$_2$ or if the total energy demand is higher. The abundant metals rank better, whereas the rank drops for the rather rare metals. If a MOF scores poorly on

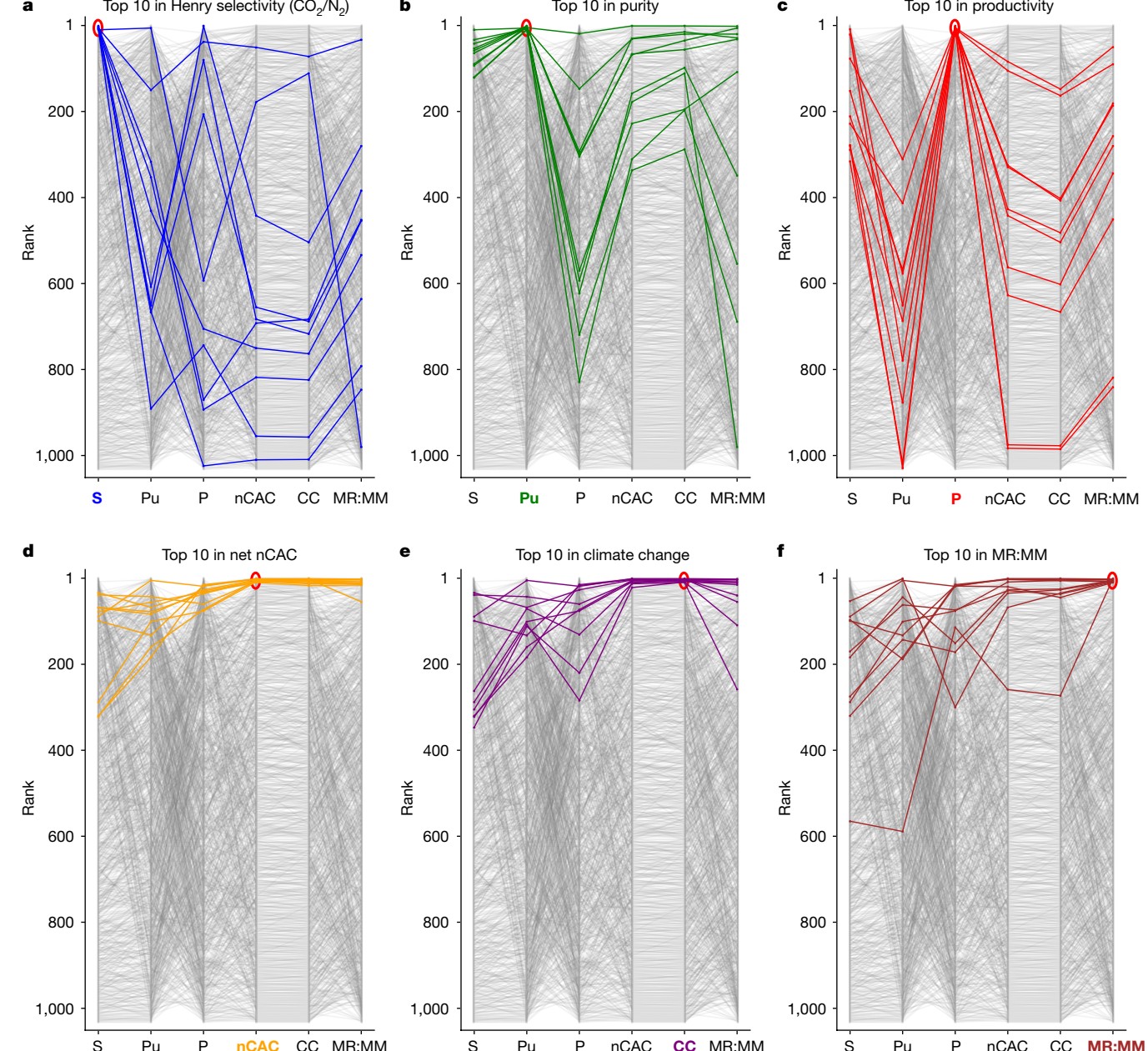

**Fig. 3 | Comparison of materials ranking for a TVSA carbon-capture process at 0.6 bar added to a cement plant in the UK.** Rankings according to Henry selectivity (S), purity (Pu), productivity (P), nCAC, climate change (CC) and MR:MM for a TVSA carbon-capture process added to a cement plant in the UK. In these graphs, the top-performing material is ranked number one. Coloured lines represent the top ten performers for the six reference KPIs. The same colour is used to highlight the KPI of interest. Every line illustrates how the ranking of a specific material (*y* axis) changes across all other KPIs (*x* axis). Our visualization tool[25] gives an interactive version of this graph.

MR:MM, it may inspire chemists to explore similar structures with more abundant metals.

Another important factor in MOF synthesis is solvent selection. The PrISMa platform identifies the greenest solvent from a list of frequently used ones. Supplementary Information Section 8.2.3 pinpoints anticipated environmental hotspots related to solvent selection.

The platform provides additional KPIs related to the process's environmental impacts (Extended Data Fig. 2b), for example, impact on ecosystem quality, human health and the use of resources (land, water, materials and non-renewable energy), and allows us to flag materials that impact the environment.

The CO$_2$-producer perspective seeks the most cost-effective capture technology. For instance, a cement producer can select different utilities based on their impact on the plant's environmental footprint and cost. In Switzerland, CO$_2$e emissions can be reduced using electric boilers instead of natural-gas-fired ones. This change significantly lowers the climate change KPI owing to the low carbon intensity of Switzerland's electricity grid, resulting in nearly 100% effective recovery. However, this improvement comes with a cost increase of approximately €16 tCO$_2^{-1}$ (about 20%) owing to the high operating costs in Switzerland (Supplementary Information Section 8.2.2).

If one needs to perform large-scale carbon capture tomorrow, the default choice is often the well-established MEA technology. However, from an investor's perspective, our platform shows that solid-sorbent-based capture processes can outperform the MEA benchmark. The cost reductions increase with CO$_2$ concentration;

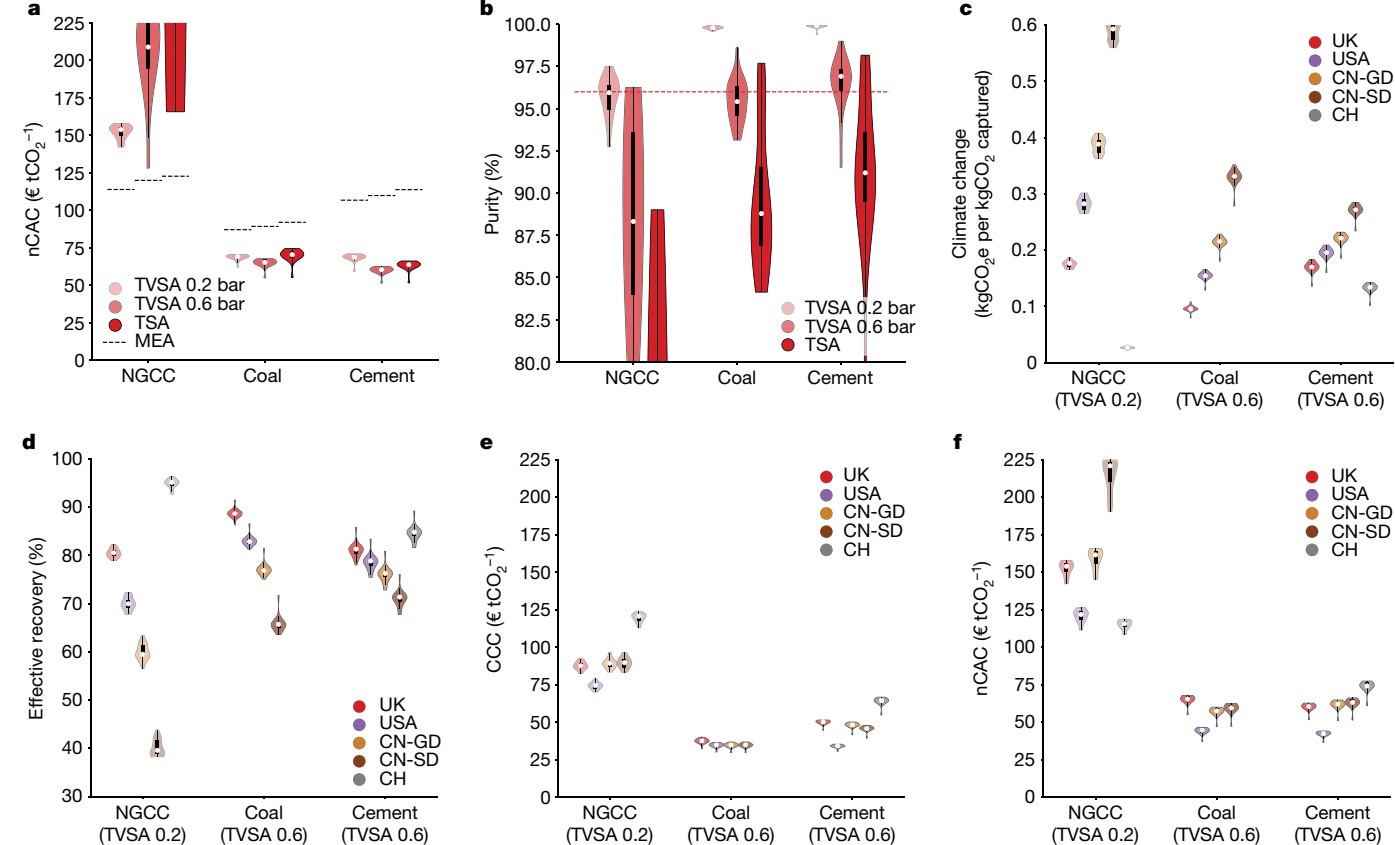

**Fig. 4 | Comparison of process configurations and regions.** This analysis of the stakeholders' perspectives focuses on the 20 materials with the lowest nCAC. In these violin plots, the white circle gives the median, which we use as a (conservative) estimate of the performance. The bottom of the violin represents a few materials with an even better performance. The width indicates the number of structures with a particular *y* value and the thick black bar contains 50% of the structures. **a**, The nCAC jointly with the MEA benchmark (black dashed lines). **b**, The purity for three $CO_2$ sources depending on the technologies (TSA and TVSA with two vacuum levels 0.2 bar and 0.6 bar) jointly with the required purity of the $CO_2$ sink (red dashed line). **c**,**d**, The climate change (**c**) and effective recovery (**d**) for the five regions. **e**,**f**, The CCC (**e**) and the nCAC (**f**) for the five regions. CN-GD, China Guangdong region; CN-SD, China Shandong region; CH, Switzerland. See Supplementary Information Section 8 for the data.

for cement, the nCAC is about a factor of two lower than the benchmark (Fig. 4a). Investors are also interested in understanding the economics of deploying carbon-capture plants in different parts of the globe. The large cost differences and electricity grid characteristics will make specific regions economically more beneficial than others. Figure 4e highlights this region's dependence on the carbon-capture cost (CCC). For the cement case, electricity and natural-gas costs are low in the USA, which makes it favourable in CCC, whereas it is highest in Switzerland. The region dependency of coal costs is rather small, whereas for natural gas, it is more substantial.

However, the CCC does not account for the $CO_2$e emissions associated with operating the carbon-capture plant and the product loss (for example, electricity). The nCAC corrects the system-based CCC by the climate change KPI (Fig. 4f). The largest impact is observed in the NGCC case. The high $CO_2$e emissions of the electricity grid owing to the many coal power plants in China, particularly in Shandong province, lead to the highest nCAC. In contrast, Switzerland has the lowest because its grid is dominated by hydroelectricity. The low energy cost and $CO_2$e emissions of the electricity grid mix make the USA beneficial for coal and cement.

The route from the first synthesis of a new material to its implementation into a commercial process can take many years. It is, therefore, important, from a chemist's perspective, to provide some guidance on how molecular characteristics impact the material's performance at the very early material's design stage. An interesting practical question is whether one can synthesize materials that work well for any $CO_2$

source. Extended Data Fig. 4a compares the nCAC ranking for NGCC, coal-fired power plants and cement plants. We observe a significant change in ranking when we go from the NGCC to coal. The changes are smaller but still considerable when we move from coal to cement. This indicates the need for tailored materials for different capture applications (see Supplementary Information Section 8.5.1 for more details).

Extended Data Fig. 4b shows the increase in nCAC with wet versus dry flue gases. As the value of $\alpha$, indicating water penetration in the bed, increases, costs rise substantially, following exponential trends after a certain threshold. For cement, the increase in nCAC is at least €5.0 $tCO_2^{-1}$ (8%), and €26.7 $tCO_2^{-1}$ (22%) for NGCC. This underscores the necessity of managing moisture at lower feed-$CO_2$ partial pressures to maintain cost competitiveness. In Supplementary Information Section 9, we discuss the limits of our (ideal) model. Under non-ideal mass-transfer conditions, about 60–70% of the materials remain top performers. However, for materials with high water affinity (for example, zeolite 13X), moisture slippage into the dry part of the bed can significantly undermine their capacity and shift their ranking.

Screening more than a thousand materials enables us to use data-driven methods to identify the molecular characteristics of the top-performing materials. For cement, we demonstrate that by retaining the descriptor related to pore geometry (that is, persistence images), we can accurately predict whether a material has a lower nCAC than the MEA benchmark (Supplementary Information Section 8.5.3). These persistence images also rank the importance of each atom in these predictions, with the collection of these atoms characterizing

# Article

the molecular features that define the adsorbaphore[23]. A common feature among materials outperforming MEA is a geometrical rod of metal atoms (highlighted in Extended Data Fig. 5). These features are often associated with stacked delocalized systems (aromatic rings) separated by 6 Å to 11 Å (see also Supplementary Fig. 62).

## Extending the chemical design space

Identifying more top-performing materials increases the likelihood of advancing some to the next technological readiness level. We use density functional theory and molecular simulations to predict material properties. Although these predictions are accurate and applicable across case studies, they require substantial central-processing-unit resources and do not scale to millions of materials. To address this, we leverage platform outcomes and implement a machine-learning feedback loop to screen a much larger chemical design space.

Our machine-learning model uses the crystal structure to predict whether a material yields an nCAC above or below a given threshold. We have limited top-performing materials, so we perform the training in steps. We start using an nCAC threshold corresponding to the MEA benchmark and use this model to screen a larger database. The most promising materials are added to the platform (round 1, in Extended Data Fig. 6a). We now have more top-performing materials, which allows us to retrain the model with a lower threshold and perform the next round. Extended Data Fig. 6a,c shows that in each round, we decrease the average nCAC. Extended Data Fig. 6d–f shows the evolution of the predictions of our machine-learning model in the chemical design space. Interestingly, there is not one single cluster of top-performing materials but several clusters of chemically different materials. This model needs to be trained for each case study. Indeed, Extended Data Fig. 6b shows that the added top-performing materials for cement do not similarly reduce the nCAC for the NGCC case.

## Experimental testing

The impact of our in silico screening is limited if it cannot reflect the experimental performance of the material. As an example, we uploaded the crystal structure of a new material, MIP-212 (Extended Data Fig. 7a). Extended Data Fig. 8 shows this is a promising material, and we studied the performance in detail (Supplementary Information Section 12.1). The experimental breakthrough curves (Extended Data Fig. 7b) show the separation between the column's predicted wet and dry fronts. The significant lapse between the breakthrough times of $CO_2$ and $H_2O$ indicates moisture penetration below 5% of the bed length, which is in good agreement with our predictions (Supplementary Information Section 12.1.3).

We also ranked CALF-20 in Extended Data Fig. 8, which gives an nCAC of €72 $tCO_2^{-1}$.

CALF-20 is being commercialized, and the estimated $CO_2$ capture cost for the Svante process is €50 $tCO_2^{-1}$ (ref. 24). A head-to-head comparison is, however, difficult as the two processes fundamentally differ.

## Outlook

The PrISMa platform's holistic approach identifies promising sorbent materials for carbon-capture applications. This modular platform extends beyond carbon capture, allowing for additional modules, for example, other gas separations and hydrogen or methane storage. Bridging fundamental research and large-scale deployment accelerates the successful implementation of innovations.

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

## Data availability

All the results obtained by the platform for all case studies presented in this work have been deposited on Zenodo at https://doi.org/10.5281/zenodo.11244258 (ref. 26). On this website, one can also find the crystal structure (cif files) of all materials studied in this work, together with the simulated isotherms, values of the heat capacity and the data that characterize the materials. The results of this work can also be accessed through our visualization tool on the Materials Cloud at https://prisma.materialscloud.io/ (ref. 25). This tool allows users to inspect all case studies and all KPIs. In addition, the Materials Cloud provides interactive versions of the graphs presented in this work. Updates and new case studies will be made available through the Materials Cloud. This tool also allows the crystal structure of materials to be uploaded and analysed across various case studies.

## Code availability

The code for the analysis of the persistence images and the interactive visualization tool can be found at https://github.com/kjappelbaum/prisma-adosorbaphore and https://github.com/ElMouba/PrISMa_VisTool, respectively. The software to run the different layers of the platform can be obtained from the corresponding authors upon request.

26. Charalambous, C. et al. A holistic platform for accelerating sorbent-based carbon capture. *Zenodo* https://doi.org/10.5281/zenodo.11244258 (2024).

**Acknowledgements** We thank M. Pougin, S. Majumdar, E. García-Díez, V. Kulakova, B. Winter and F. de Meyer for contributing to this work; and H. Celik, R. Giovine and UC Berkeley's NMR facility in the College of Chemistry (CoC-NMR) for spectroscopic assistance. We also thank I. Dovgaliuk for helping us in solving the structure of MIP-212. This work is part of the PrISMa Project (299659), funded through the ACT Programme (Accelerating CCS Technologies, Horizon 2020 Project 294766). Financial contributions from the Department for Business, Energy and Industrial Strategy (BEIS), together with extra funding from the NERC and EPSRC research councils, United Kingdom, the Research Council of Norway (RCN), the Swiss Federal Office of Energy (SFOE), and the US Department of Energy are gratefully acknowledged. Additional financial support from TOTAL and Equinor is also gratefully acknowledged. We also acknowledge funding from the USorb-DAC Project, supported by a grant from The Grantham Foundation for the Protection of the Environment to RMI's climate tech accelerator programme, Third Derivative. C.C., J.-Y.W., L.H. and S.G. are also supported by the UKRI ISCF Industrial Challenge within the UK Industrial Decarbonisation Research and Innovation Centre (IDRIC) award number EP/V027050/1. The NMR instrument used in this work is supported by the National Science Foundation under grant number 2018784. We also gratefully acknowledge Solverlo Ltd's contribution to developing the TEA module.

**Author contributions** E.M., S.M.M., K.M.J., J.V.H. and B.S. developed the materials layer. C.C., J.-Y.W., F.M., S.B.P. and S.G. developed the process layer. E.S.F., C.C., L.H. and S.G. developed the techno-economic analysis layer. J.S. and A.B. developed the life-cycle assessment layer and optimization framework. F.M. and K.M.J. developed the machine-learning methods. E.M. and J.V.H. developed the visualization software. The MIP-212 MOF was synthesized and characterized by G.M., C.S., A.Y.O., A.P., A.-Y.S. and J.A.R., and breakthrough curves were conducted and analysed by M.G., F.M., S.B.P. and S.G. All authors contributed to analysing the data and writing the paper.

**Competing interests** The authors declare no competing interests.

**Additional information**
**Correspondence and requests for materials** should be addressed to Berend Smit or Susana Garcia.

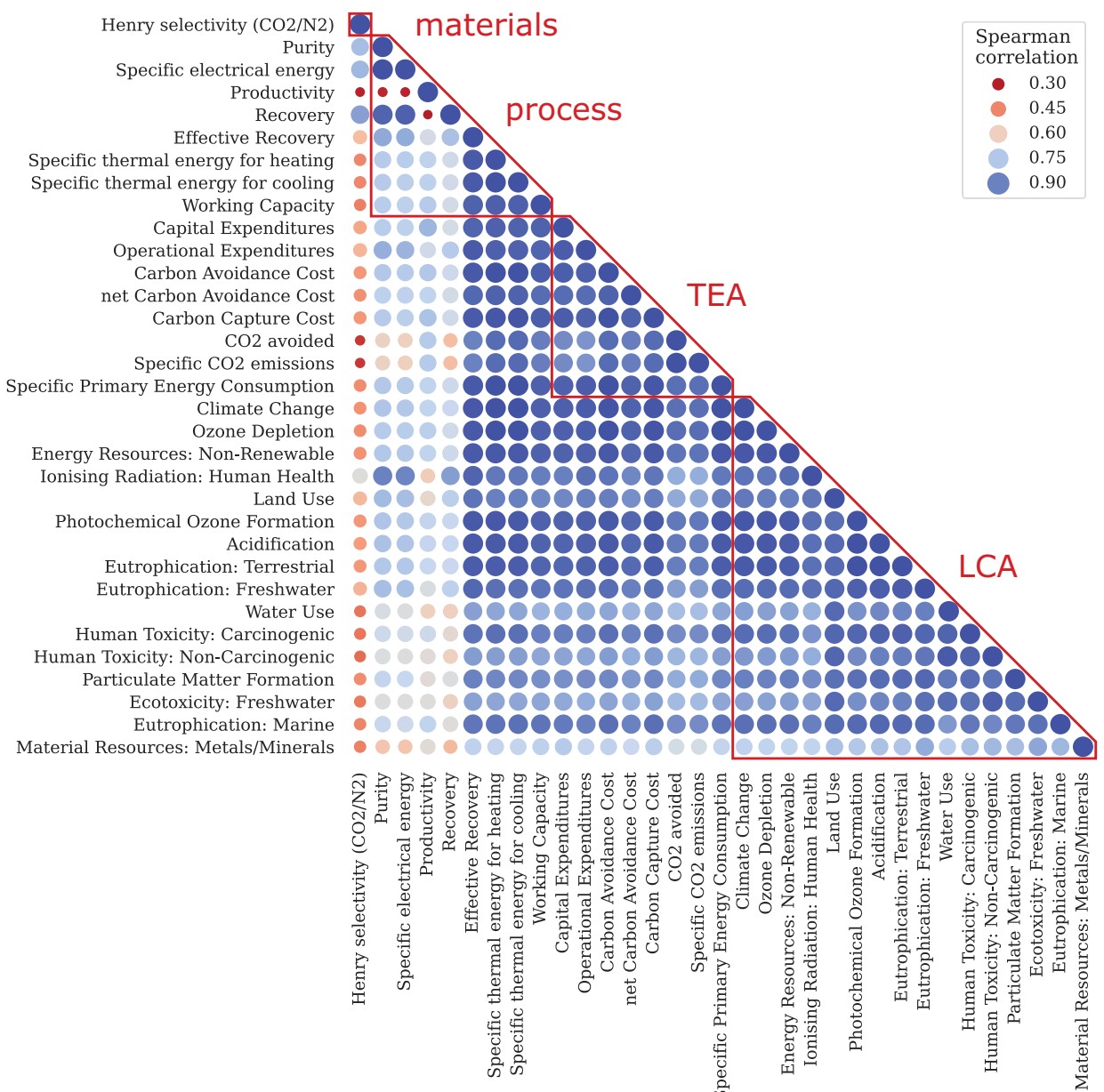

**Extended Data Fig. 1 | Spearman's rank correlation matrix for the cement case in the UK with TVSA process at 0.6 bar.** Spearman's rank correlation matrix of the rankings considering one material KPI, eight process KPIs, eight TEA KPIs, and 16 LCA KPIs. Dark blue represents very strong correlations, while dark red represents lower correlations. The size of the circle is proportional to the absolute value of the correlation. The diagonal circles in the matrix have, by definition, a Spearman's correlation coefficient of 1. A more detailed description can be found in Supplementary Information Section 7.

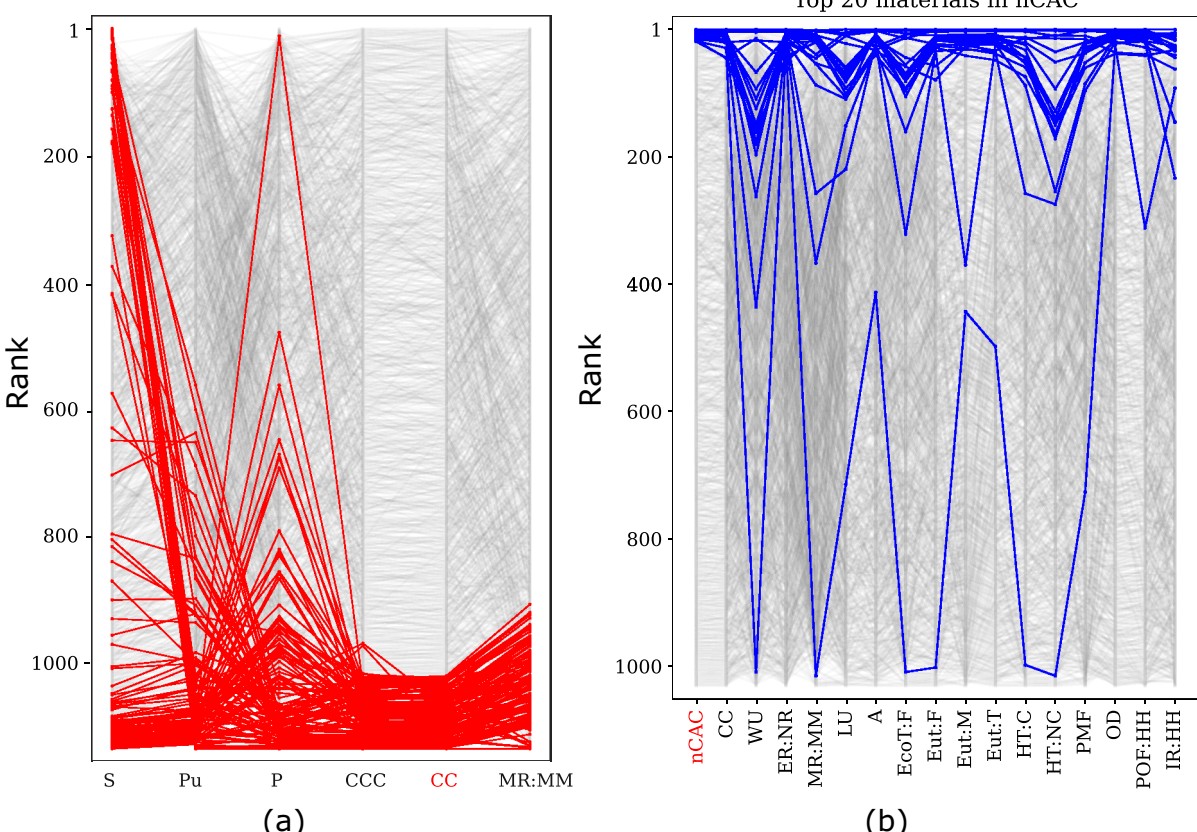

(a)

Top 20 materials in nCAC

(b)

**Extended Data Fig. 2 | Materials ranking for LCA-KPIs for cement in the UK with TVSA process at 0.6 bar.** (a) The materials that are colored red have a Climate Change KPI larger than 1, which implies that the total $CO_2$-eq. emissions of the capture plant using this material are larger than the amount of $CO_2$ that is captured. Note that we changed nCAC to the Carbon Capture Cost (CCC) (see Supplementary Information Section 6.3.6) because the nCAC is not defined for these materials. (b) Material ranking for all 16 main LCA KPIs: Climate Change (CC), Water Use (WU), Energy Resources: Non-Renewable (ER:NR), Material Resources: Metals/Minerals (MR:MM), Land Use (LU), Acidification (A),

Ecotoxicity: Freshwater (EcoT:F), Eutrophication: Freshwater (Eut:F), Eutrophication: Marine (Eut:M), Eutrophication: Terrestrial (Eut:T), Human Toxicity: Carcinogenic (HT:C), Human Toxicity: Non-Carcinogenic (HT:NC), Particulate Matter Formation (PMF), Ozone Depletion (OD), Photochemical Ozone Formation: Human Health (POF:HH), and Ionising Radiation: Human Health (IR:HH). The blue lines show the top 20 materials for the nCAC; the one material showing significant environmental hotspots in 10/16 impact categories contains silver.

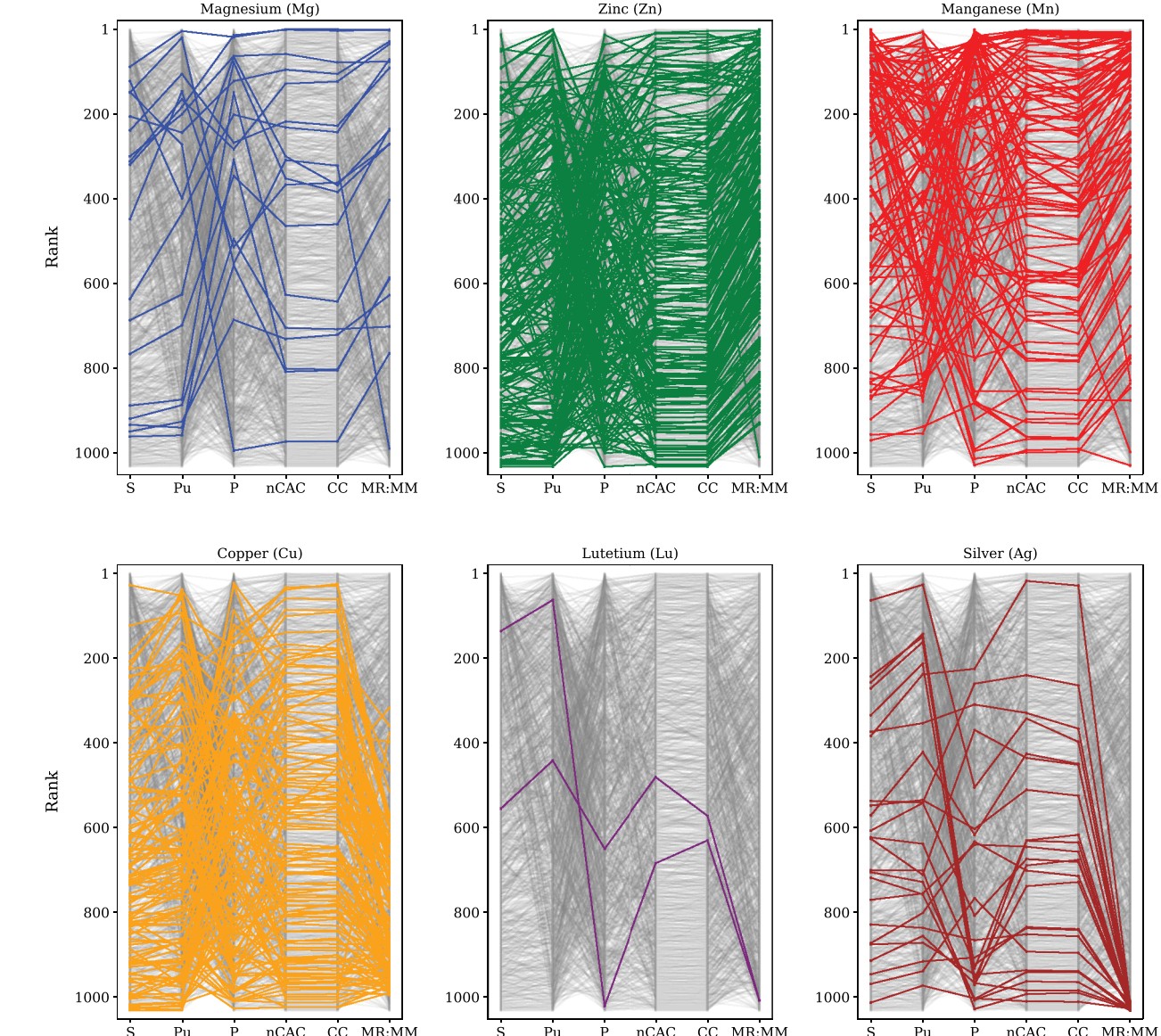

**Extended Data Fig. 3 | Ranking of the two classes of metals for the cement case in the UK with TVSA process at 0.6 bar.** Top: abundant metals (Mg, Zn, Mn), and Bottom: more rare metals (Cu, Lu, Ag). Some MOFs contain more than one type of metal. All these metals are considered in the KPI MR:MM and can lower the ranks significantly. A combination of two or three metals is, for example, contained in the worst-performing Manganese (Mn) materials, leading to their bad performance in MR:MM compared to the other materials containing the same metal.

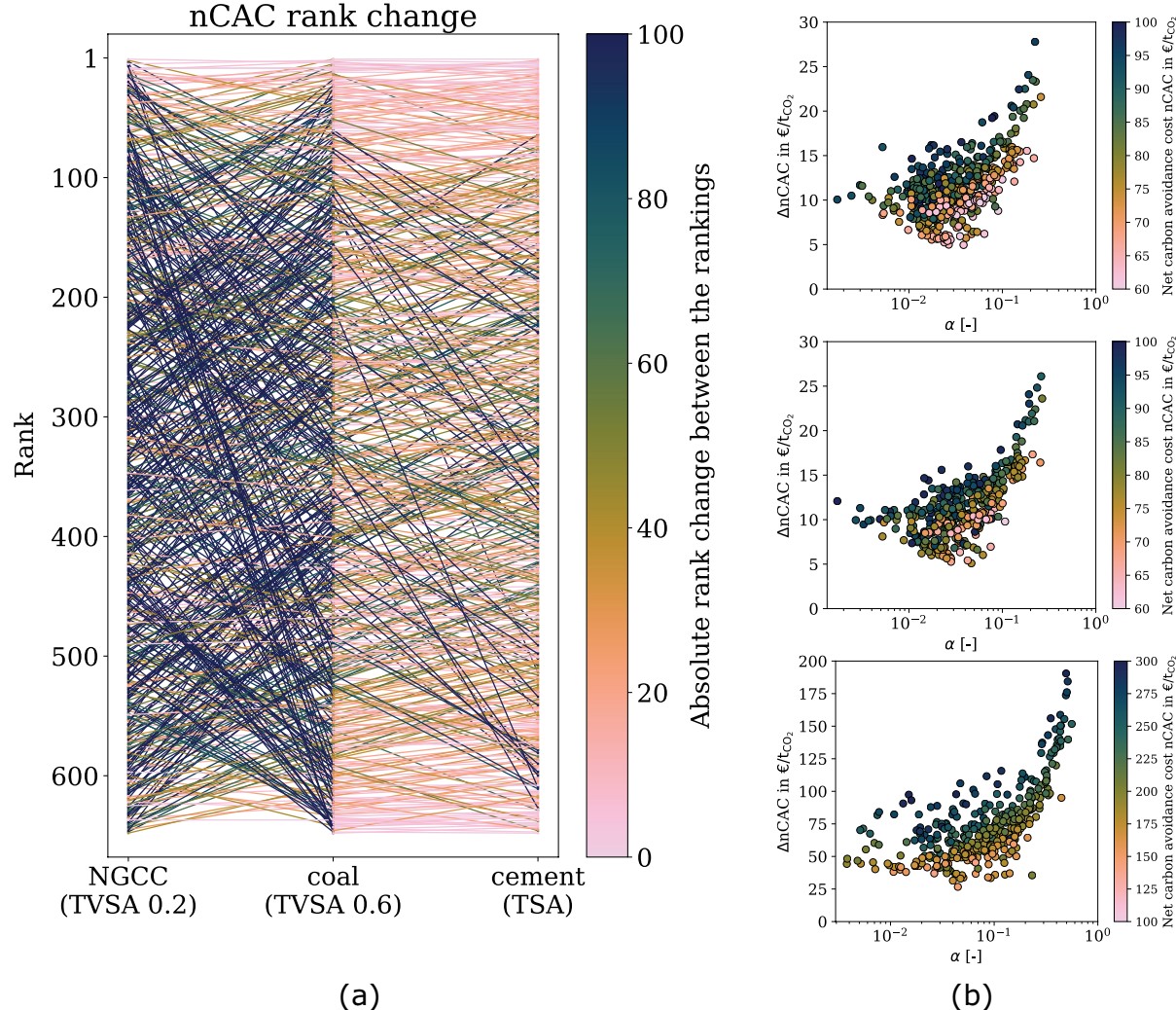

(a)

(b)

**Extended Data Fig. 4 | Ranking of materials and comparison between wet and dry flue gasses.** (a) Ranking of materials for NGCC power plant (TVSA at 0.2 bar), coal power plant (TVSA at 0.6 bar), and cement plant (TVSA at 0.6 bar). The materials are ranked using the preferred technology according to the Net Carbon Avoidance Cost (nCAC). The color coding of the lines shows the number of ranks a material change ranking. (b) Scatter plots of the increase in nCAC from dry to wet conditions as a function of the fraction of bed that is moisture-loaded. (top) Cement plant (TVSA at 0.6 bar), (middle) coal power plant (TVSA at 0.6 bar), (bottom) NGCC power plant (TVSA at 0.2 bar).

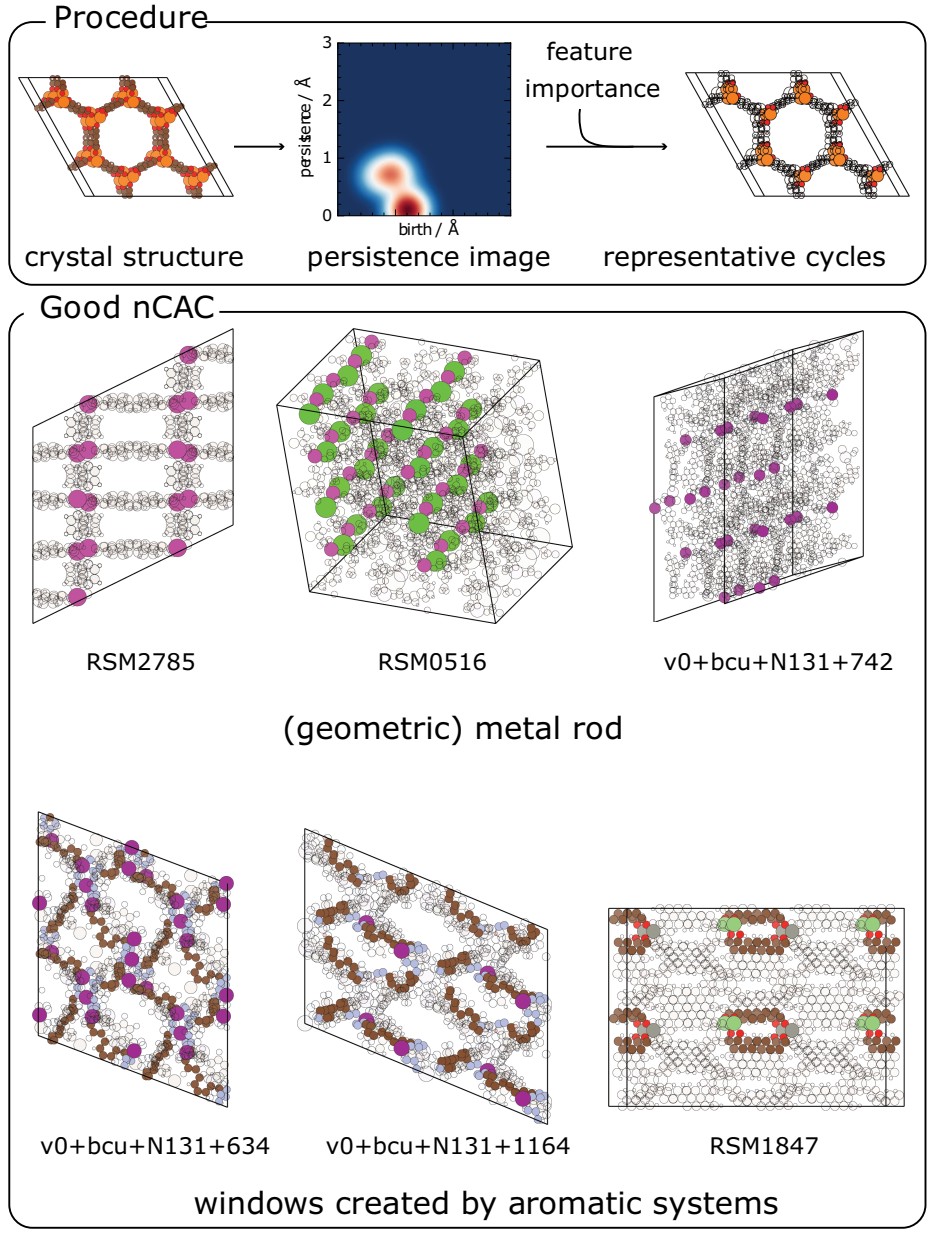

**Extended Data Fig. 5 | Identifying the adsorbaphore for the cement Case Study (TVSA, 0.6 bar) in the UK.** The top figures illustrate the methodology; the crystal structure is converted into a persistence image. We extract the most relevant pixels of the persistence images from a model trained to predict whether the nCAC is lower than the MEA-based benchmark. We then identify representative cycles, which are collections of atoms that generate a corresponding topological feature (i.e., birth/persistence pair). The bottom figure shows examples of the top-performing structures' recurring molecular features (adsorbaphores). Supplementary Information Section 8.5.3 provides more examples of these top-performing structures, and it gives the details of the methods that are used.

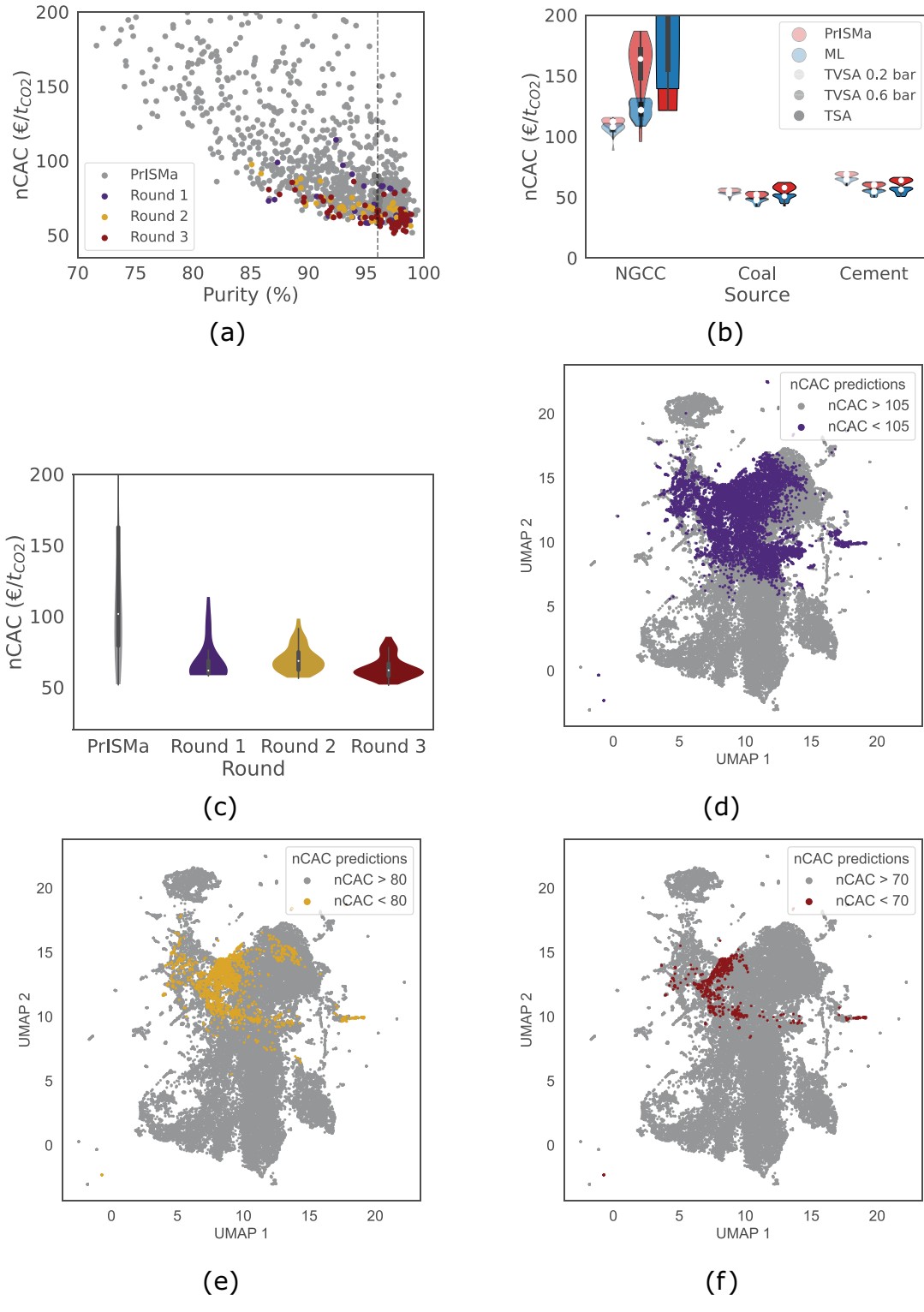

(a)

(b)

(c)

(d)

(e)

(f)

**Extended Data Fig. 6 | Iterative material discovery.** The PrISMa platform was used to train a series of ML models to predict an nCAC below a set of thresholds (105, 80, and 70 €t$^{-1}_{CO2}$) for the cement Case Study in the UK (TVSA, 0.6 bar). (a) shows the nCAC versus purity for the different rounds. (b) shows how these materials perform for the other Case Studies. (c) shows the distribution of the nCAC per round. (d)–(f) visualize the screening of the chemical space through dimensionality reduction (UMAP embedding, see Supplementary Information Section 11). Each data point corresponds to one MOF. In (d)-(f), the colored dots are MOFs with an nCAC better than the threshold and the 30,000 grey dots are from the large database.

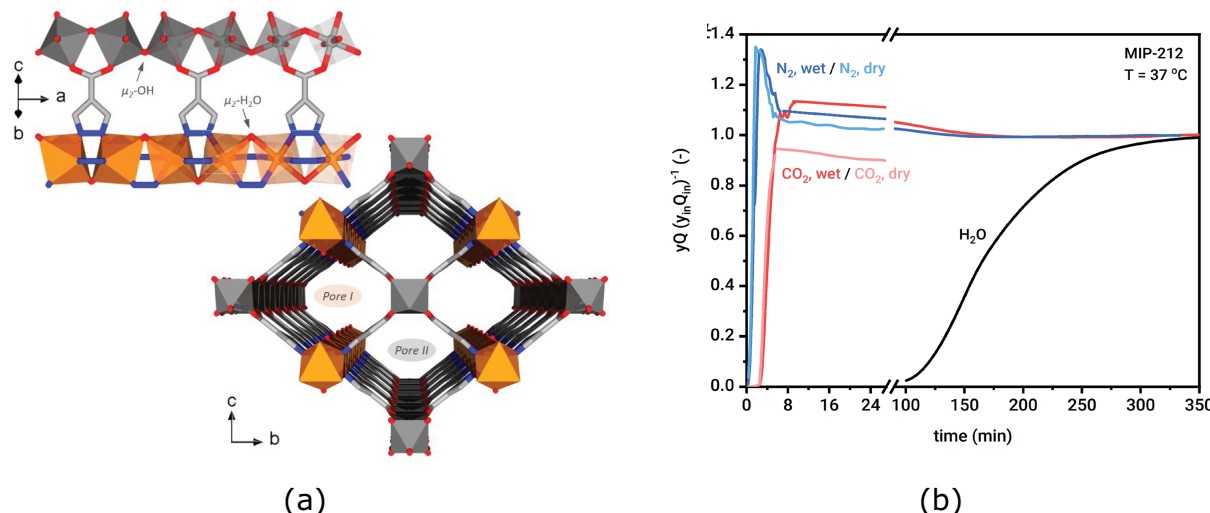

(a)

(b)

**Extended Data Fig. 7 | MIP-212.** Panel (a) shows the structure of MIP-212. MIP-212 has 1D channels constructed from an alternation of chains of Al hydroxy-carboxylates and Cu pyrazolates. Panel (b) shows the breakthrough curve under a dry and wet flue gas and conditions corresponding to the cement Case Study.

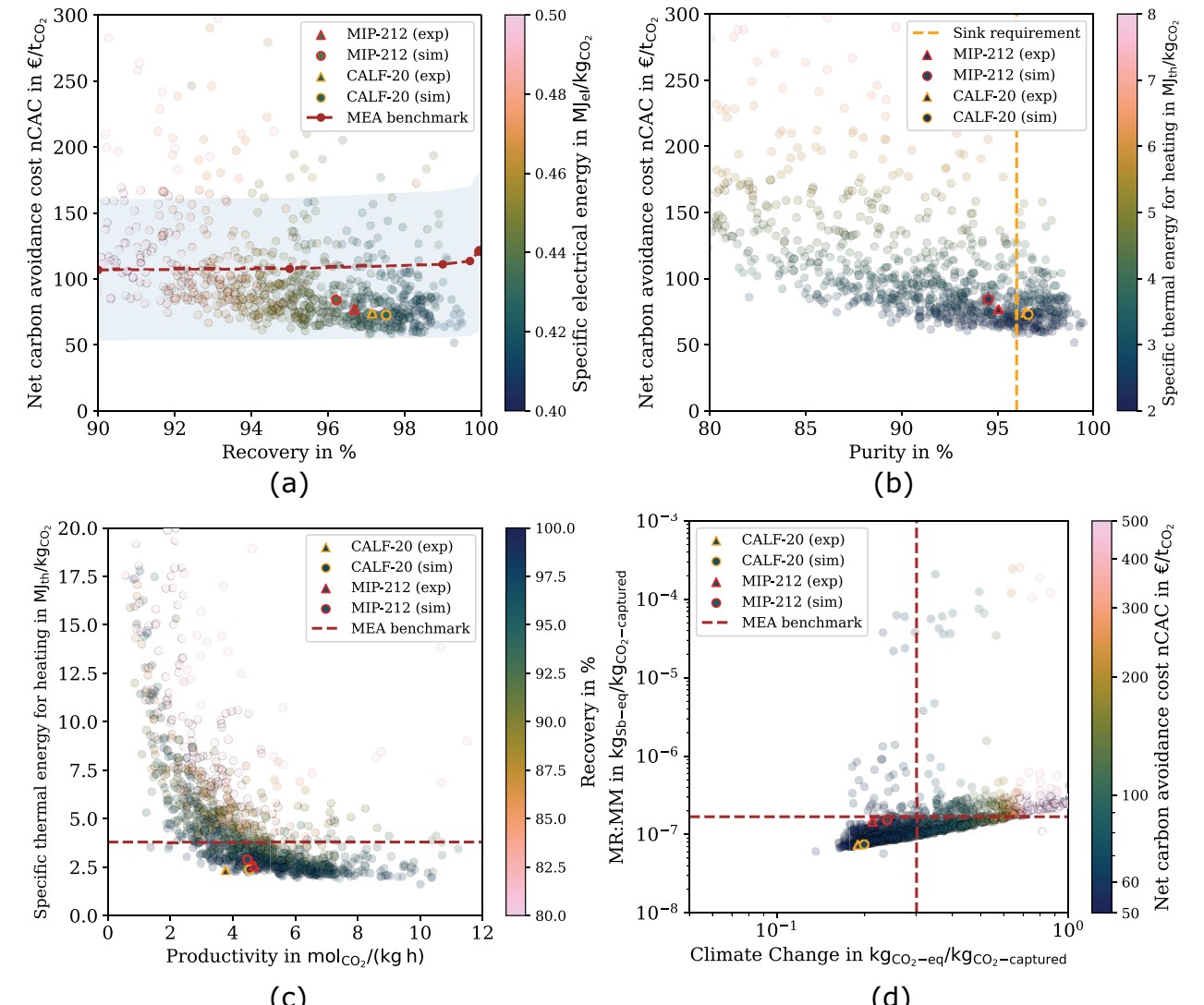

**Extended Data Fig. 8 | Materials performance for a TVSA carbon capture process at 0.6 bar added to a cement plant in the UK using experimental data.** The dotted lines in (a), (c), and (d) show the MEA benchmark, in (b), the vertical orange dotted line gives the purity required for geological storage (> 96 %), and in (a), the blue-shaded area gives the uncertainty. Each dot represents the corresponding KPI of a material. The triangles are the structures for which experimental property data is used directly in the platform (see Supplementary Information Section 12). (a) Net Carbon Avoidance Cost (nCAC) versus recovery (R) with color coding the specific electrical energy consumption, (b) nCAC versus purity (Pu) with color coding the specific thermal energy consumption, (c) Specific thermal energy consumption for heating versus productivity (P) with color coding the recovery, and (d) Material Resources: Metals/Minerals (MR:MM) versus Climate Change (CC) with color coding the nCAC.

**Extended Data Table 1 | Available case studies in the PrISMa platform**

| Source | Sink | Technology | Utility | Region |
|--------|------|-----------|---------|--------|
| NGCC<br>Coal<br>Cement | Geological storage | TSA<br>TVSA - 0.6 bar<br>TVSA - 0.2 bar | Natural gas boiler<br>Electric boiler<br>From host plant | UK<br>US<br>China (Guangdong)<br>China (Shandong)<br>Switzerland (CH) |

Any source, sink, technology, utility, and region in the platform can be combined. Switzerland has no coal-fired power plants, so this combination of source and region is not considered. Three technologies are available: temperature swing adsorption (TSA) and temperature vacuum swing adsorption (TVSA) with two vacuum levels, 0.2 and 0.6 bar. We have a total of 66 Case Studies. The complete input parameters defining these Case Studies are in Supplementary Information Table S2.

**Extended Data Table 2 | The six reference key performance indicators (KPIs)**

| KPI | Description | Definition (SI) |
|---|---|---|
| | **Materials layer** | |
| S | Ratio of the $CO_2$ and $N_2$ Henry's coefficients. | (6.1.2) |
| | **Process layer** | |
| Pu | The molar fraction of $CO_2$ in the product stream. | (6.2.1) |
| P | The amount of captured $CO_2$ per kg adsorbent during a process cycle. | (6.2.5) |
| | **Techno-economic analysis (TEA) layer** | |
| nCAC | Quantifies the cost of avoiding emitting $CO_2$ into the atmosphere over the plant's life cycle. For power generation Case Studies, the nCAC is calculated from the levelized cost of electricity and the net carbon intensity of the plant. For cement, the nCAC is calculated from the costs of carbon capture and the Climate Change (CC), as we assume that the capture plant does not affect cement production. | (6.3.4) |
| | **Life-cycle assessment (LCA) layer** | |
| CC | Gives the total Global Warming Potential (GWP) due to greenhouse gas emissions from the capture process to the air and $CO_2$ uptake from the atmosphere. | (6.4.1) |
| MR:MM | Indicates the use of non-renewable, non-fossil natural resources (i.e., minerals and metals) and considers the availability of a mineral or metal on earth and the current mining rate. The use of natural resources like minerals and metals is measured using antimony (Sb) as reference material. | (6.4.2) |

Henry selectivity (S), Purity (Pu), Productivity (P), Net Carbon Avoidance Cost (nCAC), Climate Change (CC), and Material Resources: Metals/Minerals (MR:MM). Based on Spearman analysis, we have identified six key performance indicators that describe the most important trends in each layer of the PrISMa platform (see Supplementary Information Section 7). A description of all KPIs and data generated by the platform can be found in Supplementary Information Section 6.