## [Peer Review file · Nature]

Manuscript Title: A Holistic Platform for Accelerating Sorbent-Based Carbon Capture

Reviewer Comments & Author Rebuttals

Reviewer Reports on the Initial Version:

Referees' comments:

Referee #1 (Remarks to the Author):

A. The paper presents an original assessment on the selection of MOF materials and related adsorption processes for CO₂ capture with a holistic approach based on several techno-economic-environmental KPIs.

B. The paper proposes a systematic method for the combined assessment of materials and processes in a sector where material scientists and process engineers are most of the times used to work (inefficiently) with a sequential approach rather than with an integrated one. The platform presented in this papers aims at filling this gap with an original approach.

C. Overall, the proposed methodology is sound. However, a major shortcoming (which is the main reason I believe the paper is not suitable for publication on top journal like Nature) is the assessment of the process considering dry binary CO₂-N₂ mixtures. The behaviour of MOFs with water is a key aspect in material selection and process design and its impact on costs and energy consumption cannot be neglected. Also, the method to compute the adsorption process (equilibrium based, no optimization) is not at the state of the art of PSA-TSA process analysis.

D. N/A

E. The paper provides qualitative concluding remarks, with a promotion of the PrISMa modelling platform that was developed in the project. I find the significance of the conclusions rather limited.

Other remarks:

- I believe that the title of the paper is too ambitious and too high in scope. CO₂ separation by adsorption is a very specific sector where materials and process development should really go hand in hand. In most of the other CO₂ capture technologies, the variety of materials and processes is not so wide and the proposed method would be hardly replicable and would not provide the same added value. I think that the title should reflect that authors are proposing a novel method relevant for MOF-based adsorption.

- The use of unabated natural gas boilers to supply the heat for sorbent regeneration is disputable. The same holds for the electric boiler option, which might be more efficiently substituted with a heat pump. Note that assuming a more efficient heat pump would increase the importance of an optimal selection of the TSA regeneration temperature and may affect the optimal material/process.

- From the process description, I could not find a clear statement on how the T_{med} and T_{high} temperatures are selected.

- The validation of the model against the Svante process is weak, as the Svante sorption process is very different from the one assessed in this paper.

- The equation used for the calculation of CO₂ compression power (eq. S11) is for an adiabatic compression process that does not represent properly the typical intercooled compression.

Referee #2 (Remarks to the Author):

The current paper deals with building a platform to assess sorbent-based CO₂ capture from various perspectives. The key outcome of the work is the web-based platform that allows the user to start from a molecular structure to life cycle for various sources to sink. It is written by a group of scientists who are leading experts in various aspects related to sorbent-based CO₂ capture. The paper is certainly well-written. The paper attempts to bridge scales that are several orders of magnitude apart, i.e., starting from a molecular scale and going all the way up to planetary boundaries; all in-silico. This is indeed ambitious and represents one of the holy grails in material science and engineering. Having said that, I have some major reservations in the paper being published in Nature as I have major concerns if we can experimentally validate any part of the framework. Let me elaborate

Let us start with the first layer, i.e., the prediction of isotherms. Several recent papers have highlighted some serious concerns about the accuracy of the structural information in the databases. Although the authors do mention the cleaning-up algorithms they have used, to what extent can we trust the predictions? Can we trust that all MOFs are not flexible? If they are, have they been accounted for in the simulations? Are the use of the same forcefields for the MOFs fully justified? For a number MOFs that have been synthesized and perhaps for those for which CO₂, N₂ and H₂O isotherms are available, can the authors show the comparison of their predictions with experiments (if there are multiple experimental studies on the same MOF, can we compare all of them)?

The study treats the effect of water in a rather simple manner. While the approach is certainly ok for a simple yes/no answer about the impact of water, how would it impact the process design? In many MOFs the competition of water with CO₂ is far more complex, and truth be told, we don't know how this will impact process outcomes (and hence cost)

Process layer: The TVSA cycle shown is shown in Fig S3. As explained, the authors use an equilibrium model (since no kinetic information is provided or estimated through computational methods). With the lack of such information, it would be impossible to estimate the size of the unit as the efficiency of a separator/separation is coded in the mass transfer rates. An equilibrium model can provide energy estimates but not the size of the equipment. One of the major challenges with the adsorption processes is that they need to scale better. Many scale-up studies in the literature show that owing to large volumetric flow rates, very quickly, we reach the scale-up limits, and we need to start modularizing the units at which point we lose the economies of scale. How are these factors taken into account?

Costing: The authors quote, "As there are huge uncertainties regarding the cost of MOF synthesis, our TEA model assumes the price for all MOFs to be the same (7.5 Euros per kg). How realistic is

this? My understanding is a reasonable grade zeolite 13X, which is manufactured in multi-tonne-scale, costs about 2 to 3 Euros per kg. Considering the cost of metals and those of the linkers (that are often more expensive), how realistic is this value. I have seen a recent preprint by one of the co-authors of this paper (Prof. Serre) where they estimate the large-scale cost of one of their iron MOFs to be about 30 dollars per kg. The price of CALF-20 (as stated few years ago, in an issue of global cement) is about 30 to 50 dollars per kg. Considering all of these, is it reasonable to assume 7.5 Euros per kg?

the authors claim that their estimate of a capture cost of ≈ 50 Euro per tonne is a validation of the Prisma framework. Im not quite convinced with this argument. Firstly as mentioned earlier, the cost of the CALF-20 is much higher than the value used in this work. Secondly, the process configuration Svante uses is completely different from the one considered here. Thirdly, there is no way to compare the cost breakdowns between Prisma and Svante's. In such a situation, it is my humble opinion that the proximity of the estimate is purely coincidental and is not a validation of the costing framework.

There are many other questions about the ability to synthesize an arbitrary structure meaningfully. For instance, one of the previous successes by some of the co-authors of this work was the discovery of an "adsorbaphore" (Boyd et al, Nature, 2019), which is a very interesting idea. However, the AI-PMOF has, unfortunately, to the best of my knowledge, not yet been scaled up for industrial application. What can we learn from that experience?

I want to stress that the authors have my respect for attempting a very challenging problem and I believe this is one of the best constellations of researchers to attempt this problem. We certainly need these approaches to work as they have the potential to fundamentally change the way we design materials, and I would be most happy to see them succeed. However, if I ask whether the current paper succeeds, the answer is a resounding no. My main concern is that there is far too much uncertainty in each of the layers, and there is no way to validate the framework's outcomes. Hence, I'm worried that the results, especially if published in a reputed journal such as Nature, could create unrealistic expectations and divert efforts into the wrong direction.

What is required to convince the reader that the framework works: The authors should pick a new MOF (other than CALF-20) that is predicted to be a candidate by the Prisma platform, synthesize it in the lab, show that the molecular simulations as predicted in this paper are indeed correct. Show a synthesis scheme for a kg-batch that can realistically result in a low cost of production, measure CO₂-water competitive isotherms; show long-term stability; put them in a lab-scale process using wet flue gas and confirm that they can meet targets (at the minimum CO₂ purity and capture rates).

Referee #3 (Remarks to the Author):

This manuscript addresses the problem of carbon capture by coupling multiple existing approaches

of molecular simulation of metal organic frameworks (MOFs), process analysis, techno-economic analysis (TEA), life cycle assessment (LCA). Connecting reductionist science with holistic approaches is essential for guiding research and development toward sustainability. This manuscript presents an important step in this direction by focusing on design of the large variety of MOFs while accounting for their sorption ability along with design, costs, and environmental impact. The study is very comprehensive due to the large number of MOFs considered along with the many key performance indicators. The on-line tool is a further plus. Such work is very much needed to bridge the gaps that exist between these areas of research activity. This coupling results in the PRISMA framework, which is implemented as an open source tool. The breadth of this work is impressive and the results are likely to be useful to guide the transition to a net-zero impact industry.

Several questions and concerns come to mind, as listed below.

1. Most importantly, the coupling between the four areas is loose and mostly from Step 4 to 1 in Fig. 1. Feedback from Step 1 to 4 is mostly absent, even though that is where a lot of value could be. The chemist's perspective on Page 12 gives an idea of this value, but insight can be obtained from other perspectives as well, and can then be used to modify the life cycle, economic aspects, process design, and MOFs. Greater use of systems engineering methods could have tightened the coupling and shed more light into the development of attractive MOFs for carbon capture. There is some literature on multiscale process engineering that is related to this work.

2. I felt that the main manuscript focused more on the approach and less on the insight from the analysis. Due to the loose coupling the overall approach is not particularly innovative. The resulting insight is more interesting. Some of the insight is presented in the main manuscript, mainly in the "chemists perspective". Other insight, like that from the LCA, is in the SI. My suggestion is that the insight from each perspective and the combined insight from all perspectives be included in the main manuscript. Having said this and as mentioned in Point 1, I feel that the combined insight could be stronger with more advanced systems engineering approaches.

3. For covering the breadth of methods, the approach relies on some important assumptions, one of them being the assumption that water can be used as a solvent. This assumption can have a large impact on the results, so some discussion about the validity of this assumption is needed. Analysis of the effect of this and other assumptions would be nice, as is commonly done in LCA. It can provide useful insight. Other such assumptions are the vacuum levels and CO₂ sources.

4. Fig 1 in the main paper looks nice but it is too general, and more like a graphical abstract. A figure with more details like Fig. S1 will be more useful.

5. As demonstrated in Fig. 1 and Section 4 of the supplementary material, your study includes the scale-up from materials to process-level data. However, this raises a few questions and considerations regarding the practical implementation of your findings.

- Process Operation: Since pressure swing adsorption and temperature swing adsorption are typically operated in batches using multiple beds for continuous cycles, could you please clarify if

your process-level data is also intended for continuous operation? [Refs 3 and 4]

- **MOF Replacement Rates:** the study considered the replacement cost for MOFs. However, it might be essential to think that different MOFs could have varying replacement rates due to potential differences in stability and susceptibility to damage or contamination over time. Have you taken this variability into account in your study? [Ref 5]
- **Real-world Process Discrepancies:** It's essential to acknowledge that experimentation results may differ significantly from actual process data due to various real-world complexities. Therefore, could you discuss the potential deviations when scaling up the process, considering factors like heat and electricity requirements in practical scenarios? [Refs 6 and 7]
- **Scaling Process Reliability:** In Section 8 of the supplementary material, you discussed the scaling process, but I would be interested to know more about the reliability of this scaling process for practical engineering applications. How confident are you that the scaled-up process will accurately represent the performance of the lab-scale setup? [Refs 6 and 7]

[Ref 3] Shah, Goldy, et al. "Comprehending the contemporary state of art in biogas enrichment and CO₂ capture technologies via swing adsorption." *International Journal of Hydrogen Energy* 46.9 (2021): 6588-6612.

[Ref 4] Gucuyener, Canan, et al. "Ethane/ethene separation turned on its head: selective ethane adsorption on the metal-organic framework ZIF-7 through a gate-opening mechanism." *Journal of the American Chemical Society* 132.50 (2010): 17704-17706.

[Ref 5] Balogun, Hamed A., et al. "Are we missing something when evaluating adsorbents for CO₂ capture at the system level?." *Energy & Environmental Science* 14.12 (2021): 6360-6380.

[Ref 6] Zeyde, Roman, Michael Elad, and Matan Protter. "On single image scale-up using sparse-representations." *Curves and Surfaces: 7th International Conference, Avignon, France, June 24-30, 2010, Revised Selected Papers 7*. Springer Berlin Heidelberg, 2012.

[Ref 7] Hu, Peiyu, et al. "Scale-up of open zeolite bed reactors for sorption energy storage: Theory and experiment." *Energy and Buildings* 264 (2022): 112077.

6. I wonder about the validity of claims in the abstract about "cheapest", "optimal" and "best". Can they be justified without optimization?

RESPONSE TO THE REVIEWERS

The reviewers' comments have resulted in a major improvement of the Platform, and we are grateful to them for their suggestions. In the revised manuscript, we have indicated the main changes in red. The most significant improvement is that we have incorporated water in the analysis, i.e., all our results now correspond to ternary mixtures of CO₂, N₂, and H₂O in the feed gas stream. As a result, all our graphs and discussion in the main manuscript and the Supplementary Information have been updated.

All the improvements are described in detail in the supplementary information, which is substantially revised, but a summary is presented below:

WATER INTEGRATION: In the original version, we considered dry flue gasses; in the revised version, we assessed the carbon capture process considering ternary CO₂-N₂-H₂O mixtures. The impact of water on materials and process performance, materials ranking, costs, and energy consumption is now considered and discussed. The impact that our (ideal) model assumptions have on process performance (i.e., mass transfer limitations and moisture slippage) has also been evaluated and presented.

OPTIMIZATION: In the revised version, we optimize the most relevant process and design parameters for the identified top-performing materials.

FEEDBACK LOOPS: In the revised manuscript, we describe the different feedback loops we have incorporated into the platform. In one of them, we leverage the outcomes from the platform to screen a much larger chemical design space. This feedback strategy increased the number of top-performing materials.

LCA LAYER: In the revised version, we developed a new model to predict the greenest solvent that can be selected for synthesizing a given MOF.

MODELS' VALIDATION AND ERROR PROPAGATION: We now describe how we have validated the different models in the different layers of the platform. We also analyzed the impact of uncertainty in calculating input parameters on our KPIs.

MATERIALS LAYER: In the revised version, we compare predicted versus experimental isotherms for CO₂, N₂, and H₂O (where possible) for a representative set of 18 structures.

EXPERIMENTAL TESTING: We synthesized, characterized, and tested the performance (i.e., breakthrough curves) of a novel material, MIP-212, identified as one of the top-performing ones, to validate the predictions from the platform.

We give a detailed response to the individual comments from the reviewers on the next pages.

RESPONSE TO THE REVIEWERS

REVIEWER 1

Reviewer Point P 1.1 — The paper presents an original assessment on the selection of MOF materials and related adsorption processes for CO₂ capture with a holistic approach based on several techno-economic-environmental KPIs.

The paper proposes a systematic method for the combined assessment of materials and processes in a sector where material scientists and process engineers are most of the times used to work (inefficiently) with a sequential approach rather than with an integrated one. The platform presented in this paper aims at filling this gap with an original approach.

Overall, the proposed methodology is sound. However, a major shortcoming (which is the main reason I believe the paper is not suitable for publication on top journal like Nature) is the assessment of the process considering dry binary CO₂-N₂ mixtures. The behaviour of MOFs with water is a key aspect in material selection and process design and its impact on costs and energy consumption cannot be neglected.

Reply: *The reviewer makes a good point about the importance of water.*

In the revised version, we have replaced the separation of dry binary CO₂-N₂ mixtures with ternary CO₂-H₂O-N₂ mixtures, where water is an integral part of the process design and approach. All the results presented in the main manuscript have now been obtained with this new model. A detailed description of the model can be found in SI-section 3.2.1.

The impact of water on materials and process performance (including the regeneration penalty due to water sorption/desorption and the costs due to the operation with wet gas mixtures) and materials ranking are now considered and discussed (SI-section 9). We also assess the impact of our ideal mass transfer and plug-flow assumptions on obtained nCAC values (see SI-section 9.3.2).

Reviewer Point P 1.2 — Also, the method to compute the adsorption process (equilibrium-based, no optimization) is not at the state of the art of PSA-TSA process analysis.

Reply: *In the original article, we assumed a fixed process geometry and conditions and screened for the optimal materials. Following the suggestion of the reviewer, we have added an optimization step for the top-performing materials. In particular, we have added in the revised manuscript:*

- *SI-section 10.1, in which we compute the sensitivity of the process design parameters, operating conditions, and input parameters to our TEA and LCA layers on the nCAC. From these results, we obtain so-called tornado plots that rank the parameters with the largest impact on the nCAC.*
- *SI-section 10.2 and 10.3, in which we introduce the optimization strategy used to optimize the most relevant process and design degrees of freedom, and we discuss the results.*

The optimization reduces the nCAC by about 10% for the top-performing materials (top 20) but more (20-30%) for the top 20-50. While there is very little change in the ranking of the materials for the top 20, we see a more significant change for the top 20-50.

The main reason we use an equilibrium-based model and not a more detailed one is the following: with our platform, we are screening databases that contain thousands of materials. Based on our holistic approach and integrated models, this first-order screening exercise aims to identify sets of top-performing materials for further development. These top performers can then be channeled through more detailed studies, where more accurate performance predictions can be established. Likewise, aspects like sorbent's recyclability, synthesis at the

kg scale, etc., can be investigated. State-of-the-art process models, including detailed kinetics, will be used in that second phase. For this phase, our platform limits the number of materials to only tens of them, speeding up materials discovery and guiding R&D efforts towards achievable performance targets at scale.

We added the following statement in the main paper to clarify the platform's role.

By following this holistic approach, the platform identifies sets of top-performing materials. These structures can then be funneled to follow-up studies, where more detailed process models will be used and specific aspects (e.g., sorbents durability, manufacturing) investigated to bring up the technology to pilot and demonstration scale.

Reviewer Point P 1.3 — D. N/A

Reply: *no comments provided*

Reviewer Point P 1.4 — The paper provides qualitative concluding remarks, with a promotion of the PrISMa modeling platform that was developed in the project. I find the significance of the conclusions rather limited.

Reply: *We have modified the conclusions after incorporating the feedback from the reviewer.*

Other remarks:

Reviewer Point P 1.5 — I believe that the title of the paper is too ambitious and too high in scope. CO₂ separation by adsorption is a very specific sector where materials and process development should really go hand in hand. In most of the other CO₂ capture technologies, the variety of materials and processes is not so wide and the proposed method would be hardly replicable and would not provide the same added value. I think that the title should reflect that authors are proposing a novel method relevant for MOF-based adsorption.

Reply: *We have limited the scope, and the new title now reads:*

Shedding Light on Stakeholders' Perspectives for Sorbent-based Carbon Capture

Reviewer Point P 1.6 — Using unabated natural gas boilers to supply the heat for sorbent regeneration is disputable. The same holds for the electric boiler option, which might be more efficiently substituted with a heat pump. Assuming a more efficient heat pump would increase the importance of an optimal selection of the TSA regeneration temperature and may affect the optimal material/process.

Reply: *We agree with the reviewer that using unabated natural gas boilers to supply the thermal energy for sorbent regeneration would be disputable in a full-scale post-combustion carbon capture system. The ideal scenario would require possible synergies with other CO₂-emitting processes. In this case, the natural gas boilers would be considered so that CO₂ is abated. In our study, we have considered two options, a gas boiler, and an electric boiler, as feasible options for a stand-alone cement plant, considering regional aspects (e.g., electricity grid from the region) but not possible synergies with other industrial processes. We have added a clarification to the SI explaining why we opted for a sub-optimal unabated boiler (see SI-section 8.2.2).*

The suggestion of using a heat pump is interesting. Employing heat pumps becomes a viable option when heat can be provided at relatively low temperatures (e.g., hot water at approximately 70 °C to 80 °C). Such a solution is proposed in the FEED CCS study at Hafslund Oslo Celsio plant, where heat pumps supply hot water for district heating purposes.

In our case, initial screenings in the platform showed that we need to heat our materials to higher temperatures (e.g., a minimum of 120 °C) to reach the purity required for geological storage. However, our analysis of the state-of-the-art heat pumps showed that the technology for reaching such high temperatures on an industrial scale is not ready yet. A similar conclusion was reached in the heat integration study for the Heidelberg Norcem's Brevik cement plant in Norway and the Horizon 2020 CEMCAP Project.

In the SI, we have added a detailed section giving our analysis (See SI-section 8.2.2), as we expect that for other case studies, using heat pumps might become a viable option.

Reviewer Point P 1.7 — From the process description, I could not find a clear statement on how the Tmed and Thigh temperatures are selected.

Reply: *The reviewer is correct that there was no clear statement about the selection of Tmed and Thigh in the process description section.*

In the revised manuscript, we have added the process conditions values (i.e., Tlow, Tmed, and Thigh) and the rationale for such selection to SI-section 3.2.1.

Reviewer Point P 1.8 — The validation of the model against the Svante process is weak, as the Svante sorption process is very different from the one assessed in this paper.

Reply: *The reviewer is correct that the Svante sorption process differs greatly in their column design. In the revised manuscript, we now adjusted our statement, which now reads:*

In addition, we also ranked CALF-20 in Extended Figure 6, which gives an nCAC of 71 €/tCO₂. CALF-20 is being commercialized, and the estimated CO₂ capture cost for the Svante process is 50 €/tCO₂. A head-to-head comparison is, however, difficult as the two processes fundamentally differ.

Reviewer Point P 1.9 — The equation used for the calculation of CO₂ compression power (eq. S11) is for an adiabatic compression process that does not represent properly the typical intercooled compression.

Reply: *Equation S11 (in the original manuscript) represents one compression stage's isentropic (adiabatic and reversible) work in a compression system. The new modeled CO₂ compression train consists of an integrally geared centrifugal compressor with intercooling between each single compression stage. Condensed water removal is also modeled, which happens when the gas stream is cooled below the dew temperature, allowing for a high stage pressure ratio and efficient operation. The equations based on fundamental principles used to model the polytropic compression have been implemented in the PrISMa platform, and they have now been included in SI-section 3.2.3.*

REVIEWER 2

Reviewer Point P 2.1 — The current paper deals with building a platform to assess sorbent-based CO₂ capture from various perspectives. The key outcome of the work is the web-based platform that allows the user to start from a molecular structure to life cycle for various sources to sink. It is written by a group of scientists who are leading experts in various aspects related to sorbent-based CO₂ capture. The paper is certainly well-written. The paper attempts to bridge scales that are several orders of magnitude apart, i.e., starting from a molecular scale and going all the way up to planetary boundaries; all in-silico. This is indeed ambitious and represents one of the holy grails in material science and engineering. Having said that, I have some major reservations in the paper being published in Nature as I have major concerns if we can experimentally validate any part of the framework. Let me elaborate

Let us start with the first layer, i.e., the prediction of isotherms. Several recent papers have highlighted some serious concerns about the accuracy of the structural information in the databases. Although the authors do mention the cleaning-up algorithms they have used, to what extent can we trust the predictions? Can we trust that all MOFs are not flexible? If they are, have they been accounted for in the simulations? Are the use of the same forcefields for the MOFs fully justified? For a number MOFs that have been synthesized and perhaps for those for which CO₂, N₂ and H₂O isotherms are available, can the authors show the comparison of their predictions with experiments (if there are multiple experimental studies on the same MOF, can we compare all of them)?

Reply: *We added a new SI section (SI-section 5) to address the reviewer's comment in the revised manuscript. In this section, we included:*

- *A comparison with experimental isotherms from multiple studies for CO₂, N₂ and H₂O (where available) for a representative set of 18 structures, of which new additional experimental isotherms for seven MOFs are reported. We did notice that Al-containing MOFs had a systematic overestimation of the adsorption. In the revised version, we adjusted this force field.*
- *In addition, we have analyzed how differences with experimental data in the isotherms, heat of adsorption, and heat capacity propagate in our estimate of the nCAC (see SI-section 5.6). The error propagation was based on the observation that the crystals will never be perfect experimentally, so we assume a saturation loading of 80% of the one we obtained from the simulated isotherms. In addition, a realistic estimate of the error in the Henry regime is one order of magnitude. The corresponding tornado plot is made with these assumptions and shows that the largest impact on the nCAC is about 9 €/t_{CO₂} (see SI-section 5.6).*

Additionally, we added a section in which we have studied the effect of flexibility on the adsorption properties by implementing a procedure where we perform molecular dynamic simulations to assess the impact of flexibility on adsorption properties (SI-section 3.1.3). This resulted in flagging structures for which flexibility can have a large impact on these properties (see SI-section 8.5.2).

Reviewer Point P 2.2 —

The study treats the effect of water in a rather simple manner. While the approach is certainly ok for a simple yes/no answer about the impact of water, how would it impact the process design? In many MOFs the competition of water with CO₂ is far more complex, and truth be told, we don't know how this will impact process outcomes (and hence cost)

Reply: *This point has also been raised by Reviewer 1, and we refer to our reply to point P 1.1.*

Reviewer Point P 2.3 — Process layer: The TVSA cycle shown is shown in Fig S3. As explained, the authors use an equilibrium model (since no kinetic information is provided or estimated through computational methods). With the lack of such information, it would be impossible to estimate the size of the unit as the efficiency of a separator/separation is coded in the mass transfer rates. An equilibrium model can provide energy estimates but not the size of the equipment. One of the major challenges with the adsorption processes is that they need to scale better. Many scale-up studies in the literature show that owing to large volumetric flow rates, very quickly, we reach the scale-up limits, and we need to start modularizing the units at which point we lose the economies of scale. How are these factors taken into account?

Reply: *It looks like we have not been sufficiently clear in explaining how we dimensioned our equipment:*

- *To ensure that we have a scaleable solution, the starting point in our design is a column with typical dimensions, i.e., we can make a realistic estimate of the costs.*
- *The productivity of the column follows from the cycle time. The referee is correct that we assume no mass transfer limitation, but the impact of mass transfer limitation is now analyzed in SI-section 9.3.1. The heating/cooling rates, in combination with the breakthrough times, specify the cycle time of the process.*
- *Given the process's cycle time and the material's working capacity, we scale up by determining the number of columns and trains required to process the feed flow rates from the different sources (cf. SI-section 6.2.9).*

Additionally, in the revised manuscript, we assess the impact of our ideal mass transfer assumption on obtained nCAC values, and the results and discussion are presented in SI-section 9.3.1.

Reviewer Point P 2.4 — Costing: The authors quote, “As there are huge uncertainties regarding the cost of MOF synthesis, our TEA model assumes the price for all MOFs to be the same (7.5 Euros per kg). How realistic is this? My understanding is a reasonable grade zeolite 13X, which is manufactured in multi-tonne-scale, costs about 2 to 3 Euros per kg. Considering the cost of metals and those of the linkers (that are often more expensive), how realistic is this value. I have seen a recent preprint by one of the co-authors of this paper (Prof. Serre) where they estimate the large-scale cost of one of their iron MOFs to be about 30 dollars per kg. The price of CALF-20 (as stated few years ago, in an issue of global cement) is about 30 to 50 dollars per kg. Considering all of these, is it reasonable to assume 7.5 Euros per kg?

Reply: *We agree with the reviewer that our cost estimate of the MOF was on the optimistic side. In the revised version, we have now used a cost of 30 € kg⁻¹.*

We have also added an analysis of the impact of the cost of the MOF (7.5, 30, 70 € kg⁻¹) on the nCAC (see SI SI-section 8.4.3), in which we show that the price of the MOF has only a modest impact on the nCAC. In addition, the MOF price has a small impact on the materials ranking and, thus, the selection of top performers.

Reviewer Point P 2.5 — The authors claim that their estimate of a capture cost of ≈ 50 Euro per tonne is a validation of the Prisma framework. I'm not quite convinced with this argument. Firstly as mentioned earlier, the cost of the CALF-20 is much higher than the value used in this work. Secondly, the process configuration Svante uses is completely different from the one considered here. Thirdly, there is no way to compare the cost breakdowns between Prisma and Svante's. In such a situation, it

is my humble opinion that the proximity of the estimate is purely coincidental and is not a validation of the costing framework.

Reply: *This point has also been raised by Reviewer 1. Please see our reply P1.8.*

Reviewer Point P 2.6 — There are many other questions about the ability to synthesize an arbitrary structure meaningfully. For instance, one of the previous successes by some of the co-authors of this work was the discovery of an “adsorbaphore” (Boyd et al, Nature, 2019), which is a very interesting idea. However, the Al-PMOF has, unfortunately, to the best of my knowledge, not yet been scaled up for industrial application. What can we learn from that experience?

Reply: *This is indeed an interesting point. The story is that we had sent a patent application, and EPFL policy is that the procedure gets abandoned if, within two years, we cannot find a company willing to license the patent and pay for the full patent. Hence, EPFL abandoned the patent. We feel, however, that this experience does not fit in the scope of this work.*

What is more interesting and relevant to the current work presented here is that back in 2019, we used very simple metrics: working capacity and selectivity. The current study goes far beyond what we achieved previously. We can now inform our high-throughput screening by the requirements of the whole system. If we rank Al-PMOF in the platform for our base Case Study (cement plant in the UK), we obtain an nCAC of 114 €/tCO₂, which is decent but not as good as that for top-performing materials such as MIP-212. The adsorbaphore concept we developed previously has also been explored in the current study, where we obtain very interesting insights into how to guide and inform the synthesis of top-performing materials for a given carbon capture application when considering a holistic approach.

Reviewer Point P 2.7 — I want to stress that the authors have my respect for attempting a very challenging problem and I believe this is one of the best constellations of researchers to attempt this problem. We certainly need these approaches to work as they have the potential to fundamentally change the way we design materials, and I would be most happy to see them succeed. However, if I ask whether the current paper succeeds, the answer is a resounding no. My main concern is that there is far too much uncertainty in each of the layers, and there is no way to validate the framework’s outcomes. Hence, I’m worried that the results, especially if published in a reputed journal such as Nature, could create unrealistic expectations and divert efforts into the wrong direction.

Reply: *This is an important point. In the revised version, we have added details on how we have validated the results obtained from each of the layers:*

MATERIALS *We added a section in the SI with a detailed comparison with experimental data (see Point P2.1).*

PROCESS *We added in the SI a section (SI-section 3.2.4) on how the process model is validated.*

TEA *We added in the SI a section on how we calibrated our cost estimates (see SI-section 3.3.5).*

LCA *We added in the SI a section (SI-section 3.4.4) on how each individual phase within the LCA layer is validated.*

In addition, we have studied how uncertainties in the process, TEA, and LCA parameters impact the nCAC for one of the top-performing materials (see SI SI-section 10.1). The SI SI-section 5.6 presents a similar analysis for the materials layer.

Reviewer Point P 2.8 — What is required to convince the reader that the framework works: The authors should pick a new MOF (other than CALF-20) that is predicted to be a candidate by the Prisma platform, synthesize it in the lab, show that the molecular simulations as predicted in this paper are indeed correct. Show a synthesis

scheme for a kg-batch that can realistically result in a low cost of production, measure CO₂-water competitive isotherms; show long-term stability; put them in a lab-scale process using wet flue gas and confirm that they can meet targets (at the minimum CO₂ purity and capture rates).

Reply: *We thank the reviewer for this vision for the way forward. To demonstrate how the PrISMa platform supports this development, we expanded the section on MIP-212, as this is an important example of how we envision the PrISMa platform to operate. The group of Serre synthesized this novel material, and when we ranked it in our platform, we observed that it would fall within the promising materials for our case studies. We have subsequently synthesized several grams of this material to further validate our adsorption behavior predictions. We measured adsorption isotherms and corroborated that our experimental values and breakthrough curves using wet flue gas matched the theoretical predictions. These new experimental data are now included in the revised version (see SI-section 12.1).*

This is an example of how the platform can identify novel materials that perform well in process experiments for further development. Thereby, the platform speeds up materials discovery by improving materials selection for large-scale testing and guiding R&D efforts towards achievable performance targets at scale.

REVIEWER 3

Reviewer Point P 3.1 — This manuscript addresses the problem of carbon capture by coupling multiple existing approaches of molecular simulation of metal organic frameworks (MOFs), process analysis, techno-economic analysis (TEA), life cycle assessment (LCA). Connecting reductionist science with holistic approaches is essential for guiding research and development toward sustainability. This manuscript presents an important step in this direction by focusing on design of the large variety of MOFs while accounting for their sorption ability along with design, costs, and environmental impact. The study is very comprehensive due to the large number of MOFs considered along with the many key performance indicators. The on-line tool is a further plus. Such work is very much needed to bridge the gaps that exist between these areas of research activity. This coupling results in the PRISMA framework, which is implemented as an open source tool. The breadth of this work is impressive and the results are likely to be useful to guide the transition to a net-zero impact industry.

Several questions and concerns come to mind, as listed below.

1. Most importantly, the coupling between the four areas is loose and mostly from Step 4 to 1 in Fig. 1. Feedback from Step 1 to 4 is mostly absent, even though that is where a lot of value could be. The chemist's perspective on Page 12 gives an idea of this value, but insight can be obtained from other perspectives as well, and can then be used to modify the life cycle, economic aspects, process design, and MOFs. Greater use of systems engineering methods could have tightened the coupling and shed more light into the development of attractive MOFs for carbon capture. There is some literature on multiscale process engineering that is related to this work.

Reply: *The reviewer makes a good point, and we have added the following in the revised manuscript:*

- *In the original manuscript, we did have some "static" feedback. We used the platform to analyze the accuracy we needed to compute the different material properties. From this feedback, we realized we had to ensure that we used IAST to predict the mixture isotherms and obtain more accurate estimates of the heat capacity. Both improvements have been published separately. In the revised manuscript, we explicitly mention this feedback (see SI-section 11.1).*
- *The reviewer's comment motivated us to use a machine-learning approach to screen for more potentially attractive MOFs using the results we obtained from the initial set of approximately 1,000 MOFs. We have now screened over 30,000 MOFs and increased the number of promising structures significantly. This part is described in detail in SI-section 11.2.*
- *An additional feedback loop was implemented by process optimization (see SI-section 10). We thank the reviewer for the valuable suggestion, which has further strengthened the demonstration of the power of the PrISMa platform.*

Reviewer Point P 3.2 — I felt that the main manuscript focused more on the approach and less on the insight from the analysis. Due to the loose coupling the overall approach is not particularly innovative. The resulting insight is more interesting. Some of the insight is presented in the main manuscript, mainly in the "chemists perspective". Other insight, like that from the LCA, is in the SI. My suggestion is that the insight from each perspective and the combined insight from all perspectives be included in the main manuscript. Having said this and as mentioned

in Point 1, I feel that the combined insight could be stronger with more advanced systems engineering approaches.

Reply: *We have now added more insights from our analysis to the main manuscript, considering all the upgrades implemented in the platform following all reviewers' comments.*

In addition to the added feedback loops (mentioned in our reply to Point 3.1), the different extensions to our platform gave us some interesting new insights. Examples of these insights include: 1) a detailed comparison of the process performance for wet and dry flue gasses; and, 2) using the output data from the platform to further explore the chemical design space in an optimum manner.

Reviewer Point P 3.3 — For covering the breadth of methods, the approach relies on some important assumptions, one of them being the assumption that water can be used as a solvent. This assumption can have a large impact on the results, so some discussion about the validity of this assumption is needed. Analysis of the effect of this and other assumptions would be nice, as is commonly done in LCA. It can provide useful insight. Other such assumptions are the vacuum levels and CO₂ sources.

Reply: *We have extended the LCA to include the most common solvents used in MOF synthesis, not only water. We use a hierarchical model that predicts the "greenest" solvent that can be used for the synthesis. This method is described in SI-section 3.4.2.*

Regarding the other assumptions made in our different layers, we have now conducted a sensitivity analysis on all relevant parameters and identified the impact of our assumptions on the nCAC of the process (see SI SI-section 10.1) and further optimized the process (see SI-section 10). We highlight that the specifications of the CO₂ sources are not assumptions but representative values of real processes (see References in Table S2 of the SI).

Reviewer Point P 3.4 — Fig 1 in the main paper looks nice but it is too general, and more like a graphical abstract. A figure with more details like Fig. S1 will be more useful.

Reply: *The revised manuscript now contains both figures.*

Reviewer Point P 3.5 — As demonstrated in Fig. 1 and Section 4 of the supplementary material, your study includes the scale-up from materials to process-level data. However, this raises a few questions and considerations regarding the practical implementation of your findings.

Process Operation: Since pressure swing adsorption and temperature swing adsorption are typically operated in batches using multiple beds for continuous cycles, could you please clarify if your process-level data is also intended for continuous operation? [Refs 3 and 4]

[Ref 3] Shah, Goldy, et al. "Comprehending the contemporary state of art in biogas enrichment and CO₂ capture technologies via swing adsorption." *International Journal of Hydrogen Energy* 46.9 (2021): 6588-6612.

[Ref 4] Gucuyener, Canan, et al. "Ethane/ethene separation turned on its head: selective ethane adsorption on the metal-organic framework ZIF-7 through a gate-opening mechanism." *Journal of the American Chemical Society* 132.50 (2010): 17704-17706.

Reply:

We have indeed considered the scaling-up required from the process design (i.e., the operation of a single column) to the capture plant to ensure continuous operation (see SI-section 3.3.1). The basis for the scaling is described in the calculation for the amount of "CO₂ captured" (see SI-section 6.2.9). See also response to Reviewer 2.

Reviewer Point P 3.6 — MOF Replacement Rates: the study considered the replacement cost for MOFs. However, it might be essential to think that different

MOFs could have varying replacement rates due to potential differences in stability and susceptibility to damage or contamination over time. Have you taken this variability into account in your study? [Ref 5]

[Ref 5] Balogun, Hammad A., et al. "Are we missing something when evaluating adsorbents for CO₂ capture at the system level?." *Energy & Environmental Science* 14.12 (2021): 6360-6380.

Reply: *In our model, we have used a fixed replacement rate of 5 years for all MOFs. In SI-section 8.4.3, we investigated the effect of different replacement rates (2 and 7 years) on the nCAC, which gave an increase of ca. 5 €/t_{CO₂}) and a decrease of ca. 1 €/t_{CO₂}, respectively. This point is stressed in Ref [5], which is now added to the text. We are not aware of any study that estimates the replacement rate of individual MOFs.*

Reviewer Point P 3.7 — Real-world Process Discrepancies: It's essential to acknowledge that experimentation results may differ significantly from actual process data due to various real-world complexities. Therefore, could you discuss the potential deviations when scaling up the process, considering factors like heat and electricity requirements in practical scenarios? [Refs 6 and 7]

[Ref 6] Zeyde, Roman, Michael Elad, and Matan Protter. "On single image scale-up using sparse-representations." *Curves and Surfaces: 7th International Conference, Avignon, France, June 24-30, 2010, Revised Selected Papers 7*. Springer Berlin Heidelberg, 2012.

[Ref 7] Hu, Peiyu, et al. "Scale-up of open zeolite bed reactors for sorption energy storage: Theory and experiment." *Energy and Buildings* 264 (2022): 112077.

Reply:

In the revised version of the manuscript, we present how we have validated our platform against real-life data (see our reply to Point 2.7, Reviewer 2). In addition, we have added a section in which we study the impact of our assumptions on the nCAC of the process (see Tornado plots in SI-section 9.3 and 10.1).

The availability of heat and electricity and the supply of utilities in practical scenarios is considered in our holistic and integrated approach (see SI-section 3.3.4), where a utility database was developed to allow for the creation of different scenarios.

Reviewer Point P 3.8 — 6. I wonder about the validity of claims in the abstract about "cheapest", "optimal" and "best". Can they be justified without optimization?

Reply: *In the revised version, we now include optimization of the top-performing structures (see SI-section 10).*

Reviewer Reports on the First Revision:

Referees' comments:

Referee #1 (Remarks to the Author):

The authors have done a considerable amount of additional work to improve the paper according to reviewers comments.

The most important remark I had in the first review round was about working with humid gas instead of dry N₂/CO₂ mixture. This has been addressed in the new version. However, being this a key property, I think that water affinity should be part of the materials variable in addition to Henry selectivity. This should also be used to understand if water affinity has a higher Spearman correlation with the TEA indicators, as the current correlation between material and TEA is surprisingly very low.

Other minor comments:

- in the caption of Figure 2 you write that "each dot represents a material". Actually, each dot should represent the optimized process (minimum CAC?) for each material. Please, explain this better, as the KPIs in that figure are clearly process KPIs.

- At p.10-11 you have the following text: "A capture process with these materials will emit more CO₂-eq. over the plant's lifetime than the total amount of CO₂ that is captured. Interestingly, some of these materials rank high in other KPIs, again highlighting the importance of obtaining insights into all aspects of the capture process."

As this is a surprising result, please give more insights on why this happens. For instance, make an example of 1-2 materials with high KPIs and explain how they achieve positive emissions.

Referee #2 (Remarks to the Author):

Review of Chartrambolous

The authors have addressed some of the queries in this version. I appreciate that they have shown the example of a new material MIP-212 that could hold potential. However, there are some critical issues in interpreting these results that I comment below. Having reviewed the changes to the paper, my main concerns about the paper remain.

1. Most results of the paper are speculative: As I mentioned in my previous review. At the same time, the framework is a good tool for studying sorbent-based CCS solutions, most are largely speculative. There is no validation barring some of the isotherms (even they show substantial variations). The authors themselves have summarized the challenges that computational predictions are made based on single crystals while the MOFs are made as powders. The authors estimate that the difference between the crystal-based predictions and the powders will be of the order of a few percentages. There are several paper in the literature that show that defects and flexibility can indeed have more drastic effects on adsorption properties.

2. MIP-212 isotherm results: I appreciate that the authors took the effort to synthesize a new material that was identified as a potential candidate. The comparison of the CO₂ isotherms in Fig S12 shows the differences in the predictions. In fact the comparison of the experimental and predicted water isotherms is quite evident. The simulations under-predict the shape and quantities of the isotherm. Indeed, the experimental capacity of water is about 1.8 to 2x that of the prediction. This will indicate that the CO₂-water competition could be potentially under predicted by the model.

3. MIP-212 breakthrough results: Fig S91 shows the dry and wet breakthrough experiments for CO₂ on MIP-212. The results are reminiscent of a classical type-1 - type-1 competitive system (e.g. water-CO₂ competition on most zeolites, including 13X that has been discussed in the paper). The roll-up of CO₂ indicates a strong water affinity. However, when the component flow rate (plotted as Q^*y) returns to 1 when water breakthrough is complete, clearly indicates that in the presence of water CO₂ will have any capacity at all. However, in Fig. S72, MIP-212 is shown to perform better than CALF-20. I would expect MIP-212 to be close to Zeolite 13X.

4. I highlight the above case of MIP-212 to an example where predictions of process (and further systems) level assessment based on pure simulations can be quite unreliable.

5. Multiple reviewers have pointed out to the limitations of an equilibrium based design. I appreciate the steps that the authors have taken. However, a critical aspect that is still missing is that of mass transfer (for both CO₂ and H₂O). As far as I could tell, this aspect has not been accounted for except in an approximate manner (in Sec 9.3.1 of the SI). The size of the plant (vessels and associated equipment) rely on this critical parameter and the energy could be affected.

In summary, framework is useful but is built on several assumptions at each layer, which is very difficult to be validated (and has not been validated in this paper). The attempt to validate the first layer has been only modestly successful. Even predicting properties, particularly water adsorption, and CO₂-water adsorption itself has been challenging, leave alone scaling-up the synthesis and using benign and cheap solvents, stability to impurities (water, SO_x/NO_x). Computational prediction of these important practical considerations are not yet mature, and understandably not included in the current analysis.

In conclusion, the paper is not ready to be published in a journal like Nature as the scale and rigour of experimental validation is still not convincing enough that computational techniques can provide a way forward to identifying MOFs for CO₂ capture. Considering that MOFs are no longer materials of academic curiosity (even for CO₂ capture), at least at a lab-scale process validation (as I had recommended in my original review) is needed to gain confidence in the PriSMa framework. Recent examples in other areas of computational material discovery highlight that the experimental burden of proof should be high, else people might lose confidence. This is a stellar team and I wish them all the best in this challenging pursuit.

Referee #3 (Remarks to the Author):

Thanks for the revised version of this manuscript. As mentioned in the first round, the attempt to bring together the views of diverse stakeholders is ambitious and commendable. The magnitude of work that forms the basis of the manuscript is impressive. Changes in the revised manuscript such as the approach for considering various solvents and the use of machine learning make it stronger.

However, upon going through the manuscript a few times along with the huge amount supporting information, I feel that the clarity of the "take-home" message is not strong and the claims made in the conclusions are not being delivered. In particular, I am not convinced that the claim in the conclusions that ends in "... shows how these decisions are interrelated, de-risking investment and providing a common basis for identifying the joint way forward" is fully valid. The abstract says, "identifying the most cost-effective technology and optimal process configuration, revealing the molecular characteristics of top-performing sorbents, determining the best locations, and informing on environmental impacts, co-benefits, and trade-offs." I see this being done in the manuscript, but mostly as analysis. No designs that make all the decisions claimed in the quoted statements above are provided as results from this work. It seems to me that what is needed is a study that starts with the 100,000 novel MOFs and screens them by incorporating the requirements of all the stakeholders and then identify the trade-offs between the most promising MOFs from which the decision maker can choose the "best" MOF, technology, process, life cycle, and location. May be the short-length format of the article makes it difficult to effectively communicate this point along with everything else that also needs to be included.

REVIEWER 1

Reviewer Point P 1.1 — The authors have done a considerable amount of additional work to improve the paper according to reviewers comments.

The most important remark I had in the first review round was about working with humid gas instead of dry N₂/CO₂ mixture. This has been addressed in the new version. However, being this a key property, I think that water affinity should be part of the materials variable in addition to Henry selectivity. This should also be used to understand if water affinity has a higher Spearman correlation with the TEA indicators, as the current correlation between material and TEA is surprisingly very low.

Reply: *The effect of water on the overall process performance is captured by two variables: α , the water penetration length into the column, and the water resistance coefficient (WRC). These variables have complex interrelationships with other KPIs. For example, if a material has a low α , this material can have good performance irrespective of the WRC. If a material has a high WRC, the value of α is irrelevant. The Spearman analysis, however, only well-describes monotone correlations. For this reason, we did not include water-related variables in the Spearman analysis but devoted a separate section to the water impact (section 9.1 of the SI).*

In the revised version, we added a paragraph to Chapter 7 to better explain this point.

Other minor comments:

Reviewer Point P 1.2 — - in the caption of Figure 2 you write that "each dot represents a material". Actually, each dot should represent the optimized process (minimum CAC?) for each material. Please, explain this better, as the KPIs in that figure are clearly process KPIs.

Reply: *The revised version now reads:*

... each dot represents the corresponding KPI of a material. ..

Reviewer Point P 1.3 — - At p.10-11 you have the following text: "A capture process with these materials will emit more CO₂-eq. over the plant's lifetime than the total amount of CO₂ that is captured. Interestingly, some of these materials rank high in other KPIs, again highlighting the importance of obtaining insights into all aspects of the capture process." As this is a surprising result, please give more insights on why this happens. For instance, make an example of 1-2 materials with high KPIs and explain how they achieve positive emissions.

Reply: *To explain this, we added to the main text:*

Interestingly, some of these materials rank high in other KPIs (e.g., Henry selectivity). There can be various reasons why using such materials can still have a high impact on climate change. For instance, some materials have a very low CO₂ working capacity, leading to a huge material and energy demand; others, even with comparatively good working capacities and moderate heat demands, contain metals like gold or rhodium. For these materials, the climate impact of their synthesis is so large that it leads to a CC higher than one. These findings again highlight the importance of obtaining insights into all aspects of the capture process.

In addition, we added a paragraph to section 8.2.3 of the SI with some additional info.

REVIEWER 2

Reviewer Point P 2.1 — The authors have addressed some of the queries in this version. I appreciate that they have shown the example of a new material MIP-212 that could hold potential. However, there are some critical issues in interpreting these results that I comment below. Having reviewed the changes to the paper, my main concerns about the paper remain.

Most results of the paper are speculative: As I mentioned in my previous review. At the same time, the framework is a good tool for studying sorbent-based CCS solutions, most are largely speculative. There is no validation barring some of the isotherms (even they show substantial variations). The authors themselves have summarized the challenges that computational predictions are made based on single crystals while the MOFs are made as powders. The authors estimate that the difference between the crystal-based predictions and the powders will be of the order of a few percentages. There are several paper in the literature that show that defects and flexibility can indeed have more drastic effects on adsorption properties.

Reply: *In her/his original review, the reviewer pointed out that more validation of the different layers in the platform was needed so the results could be trusted. As a response, we added for each layer a separate section in the SI describing how each of the layers was validated, as well as a section on the impact of parameter uncertainties on our platform results:*

- *Section 3.2.4 for the process layer*
- *Section 3.3.5 for the techno-economics layer*
- *Section 3.4.4 for the life cycle assessment layer*
- *Section 10.1, containing a sensitivity analysis on how parameter uncertainties propagate throughout the platform*
- *Section 12.1.3, describing the experimental performance testing of the predicted breakthrough curves for wet flue gasses in MIP-212, a MOF material identified as top-performing by our platform*

For the materials layer, we added a new Chapter (Chapter 5 of the SI), in which we presented a detailed comparison of our isotherm predictions with experimental data from multiple studies (as requested by Reviewer 2 in her/his original review) for 15 different MOFs, including novel experimental data. Based on this comparison, we concluded:

- *We assume an infinite, perfect crystal in our simulations. However, synthesizing such perfect crystals in practice is experimentally difficult, if not impossible. One can expect to have a 10-20% lower accessible pore volume.*
- *In our screening study, we assume a universal force field (UFF) with an adjustment for Al. The comparison with the experimental data shows that the error in the Henry coefficient can be as much as one order of magnitude.*

Importantly, we assessed the impact of those uncertainties in the process's nCAC (see Section 5.6). We showed an underestimation of the nCAC by at most 8€/tCO₂ (15%) for the top-performing materials.

In summary, in the previous review, the Reviewer made an important point about the validation, which is essential for the platform's credibility. However, we feel that Reviewer 2's opinion that the platform is not sufficiently validated is based on her/his perception that we only assert a few percent mismatch between experimental and computational values.

This is factually incorrect, we assumed one order of magnitude at low pressures and 20% at saturation, and that misperception appears to be this reviewer's main issue here and in Points 2.2, 2.4, and 2.6.

Reviewer Point P 2.2 — MIP-212 isotherm results: I appreciate that the authors took the effort to synthesize a new material that was identified as a potential candidate. The comparison of the CO₂ isotherms in Fig S12 shows the differences in the predictions. In fact the comparison of the experimental and predicted water isotherms is quite evident. The simulations under-predict the shape and quantities of the isotherm. Indeed, the experimental capacity of water is about 1.8 to 2x that of the prediction. This will indicate that the CO₂-water competition could be potentially under predicted by the model.

Reply: *The reviewer's point is similar to Point 2.1. The reviewer is correct that there are differences in the computed versus the experimental isotherms for MIP-212. However, as argued in Point 2.1, these differences do not have such an impact on the overall results as the reviewer assumes. The fact that for MIP-212, the process performance (see Response to Point 2.5) and the nCAC values computed from experimental data and simulations agree very well (< 10% deviation) emphasizes that our simulation sufficiently accurately predicts the adsorption data for the overall results to be trustworthy.*

Reviewer Point P 2.3 — MIP-212 breakthrough results: Fig S91 shows the dry and wet breakthrough experiments for CO₂ on MIP-212. the results are reminiscent of a classical type-1 - type-1 competitive system (e.g. water- CO₂ competition on most zeolites, including 13X that has been discussed in the paper). The roll- up of CO₂ indicates a strong water affinity. However, when the component flow rate (plotted as Q*y) returns to 1 when water breakthrough is complete, clearly indicates that in the presence of water CO₂ will *not* have any capacity at all. However, in Fig. S72, MIP-212 is shown to perform better than CALF-20. I would expect MIP-212 to be close to Zeolite 13X.

Reply: *These are indeed very interesting observations. We realize that the SI document is huge, and it can easily be missed that we have already made the same observation in Section 12.1.3. Indeed, CO₂ sorbed by MIP-212 is displaced by H₂O under breakthrough conditions. Note that CO₂ is also displaced by H₂O under the same conditions (85% relative humidity) for the CALF-20 system. Hence, this observation does not serve as a basis for distinguishing the performance of these systems. As our response to the first reviewer mentioned (Point 1.1), the correlation between 'water affinity' and process-level performance is not straightforward (see also the new paragraph in Chapter 7 of the SI). In Sections 9.3.1 and 9.3.2 of the SI, we have provided a thorough additional analysis of the systems' comparative performances (including zeolite 13X).*

Reviewer Point P 2.4 — I highlight the above case of MIP-212 to an example where predictions of process (and further systems) level assessment based on pure simulations can be quite unreliable.

Reply: *We respectfully disagree with the Reviewer's comment. For the specific case of MIP-212, we synthesized and fully characterized the material (See Section 12.1 in the SI document). In addition, we tested its dynamic performance in a fixed-bed reactor system. These results show good agreement between the predicted and measured moisture penetration in the bed, a key parameter for the process performance.*

Reviewer Point P 2.5 — Multiple reviewers have pointed out to the limitations of an equilibrium based design. I appreciate the steps that the authors have taken. However, a critical aspect that is still missing is that of mass transfer (for both CO₂ and H₂O). As far as I could tell, this aspect has not been accounted for except

in an approximate manner (in Sec 9.3.1 of the SI). The size of the plant (vessels and associated equipment) rely on this critical parameter and the energy could be affected.

Reply:

The reviewer remarks on the “criticality” of mass transfer, implying that the present approximate treatment is inadequate. We respectfully disagree. First, it is important to distinguish the CO₂ capture application investigated here, which is an equilibrium-driven separation, from a kinetically controlled separation process —e.g., air separation, where differing rates of mass transfer are exploited for selectivity.^{1,2}

In our data curation approach to selecting our materials database (see Section 3.1.1), we only have materials for which the limiting pore sizes surpass the largest adsorbate, N₂, i.e., all the adsorbate molecules can enter the pores freely and diffusion in the crystal is not the limiting step. This is the case for known reference materials for CO₂ capture (e.g., 13X and UTSA-16^{3,4}). Hence, we do not consider operating regimes for which ignoring the mass transfer may critically undermine predictive accuracy.

Additionally, we did consider the influence of mass transfer on various KPIs, including plant size, energy consumption, and cost. We have employed conservative approximations in our equilibrium model. In particular, we have used a weighting parameter to correct the water penetration length (see Section 9.3.1). This water penetration is the most significant variable affecting column performance for non-ideal behavior. We use previously reported experimental results of zeolite 13X for calibrating our weighting parameter, which can be seen as a worst-case scenario.

Reviewer Point P 2.6 — In summary, framework is useful but is built on several assumptions at each layer, which is very difficult to be validated (and has not been validated in this paper). The attempt to validate the first layer has been only modestly successful. Even predicting properties, particularly water adsorption, and CO₂-water adsorption itself has been challenging, leave alone scaling-up the synthesis and using benign and cheap solvents, stability to impurities (water, SO_x/NO_x). Computational prediction of these important practical considerations are not yet mature, and understandably not included in the current analysis.

In conclusion, the paper is not ready to be published in a journal like Nature as the scale and rigour of experimental validation is still not convincing enough that computational techniques can provide a way forward to identifying MOFs for CO₂ capture. Considering that MOFs are no longer materials of academic curiosity (even for CO₂ capture), at least at a lab-scale process validation (as I had recommended in my original review) is needed to gain confidence in the PrISMa framework. Recent examples in other areas of computational material discovery highlight that the experimental burden of proof should be high, else people might lose confidence. This is a stellar team and I wish them all the best in this challenging pursuit.

Reply: *We direct the Reviewer to our comment on Point 2.1, which addresses the main point of concern raised again here.*

As stated in the manuscript, the objective of the PrISMa platform is to identify, from a holistic approach, sets of promising sorbents for further development. The platform speeds up material discovery and focuses research and development (R&D) efforts towards achievable performance targets at scale. We firmly believe that we have demonstrated the reliability of our predictions towards that aim.

REVIEWER 3

Reviewer Point P 3.1 — Thanks for the revised version of this manuscript. As mentioned in the first round, the attempt to bring together the views of diverse stakeholders is ambitious and commendable. The magnitude of work that forms the basis of the manuscript is impressive. Changes in the revised manuscript such as the approach for considering various solvents and the use of machine learning make it stronger.

However, upon going through the manuscript a few times along with the huge amount supporting information, I feel that the clarity of the "take-home" message is not strong and the claims made in the conclusions are not being delivered. In particular, I am not convinced that the claim in the conclusions that ends in "... shows how these decisions are interrelated, de-risking investment and providing a common basis for identifying the joint way forward" is fully valid.

Reply: *To avoid the suggestion that we are claiming that our platform provides the ultimate answer, we slightly modified the wording by inserting "...makes the first step in providing..." This sentence now reads:*

The PrISMa platform makes the first step in providing engineers with options to identify economically and/or environmentally challenging factors in the design phase of optimal capture technologies; molecular design targets for chemists; local integration benefits for CO₂ producers; and the best locations for investors; and shows how these decisions are interrelated, de-risking investment and providing a common basis for identifying the joint way forward.

Reviewer Point P 3.2 — The abstract says, "identifying the most cost-effective technology and optimal process configuration, revealing the molecular characteristics of top-performing sorbents, determining the best locations, and informing on environmental impacts, co-benefits, and trade-offs." I see this being done in the manuscript, but mostly as analysis. No designs that make all the decisions claimed in the quoted statements above are provided as results from this work. It seems to me that what is needed is a study that starts with the 100,000 novel MOFs and screens them by incorporating the requirements of all the stakeholders and then identify the trade-offs between the most promising MOFs from which the decision maker can choose the "best" MOF, technology, process, life cycle, and location. May be the short-length format of the article makes it difficult to effectively communicate this point along with everything else that also needs to be included.

Reply: *We slightly modified the abstract to avoid making the suggestion that we are claiming to have a design that can make all the decisions. The sentence now reads:*

These studies demonstrate how the platform simultaneously informs various stakeholders: it assesses the cost-effectiveness of the technology, process configuration, and locations, reveals the molecular characteristics of top-performing sorbents and informs on environmental impacts, co-benefits, and trade-offs.

REFERENCES

- [1] Ruthven, D.; Xu, Z.; Farooq, S. *Gas Separation & Purification* **1993**, *7*, 75–81.
- [2] Farooq, S.; Ruthven, D. *Chemical Engineering Science* **1991**, *46*, 2213–2224.
- [3] Park, Y.; Ju, Y.; Park, D.; Lee, C.-H. *Chemical Engineering Journal* **2016**, *292*, 348–365.
- [4] Agueda, V. I.; Delgado, J. A.; Uguina, M. A.; Brea, P.; Spjelkavik, A. I.; Blom, R.; Grande, C. *Chemical Engineering Science* **2015**, *124*, 159–169, Metal-Organic Frameworks for Emerging Chemical Technologies.

Reviewer Reports on the Second Revision:

Referees' comments:

Referee #2 (Remarks to the Author):

I thank the authors for responding to my queries. In my review, I pointed out that while the synthesis of MIP-22 is appreciated, there were major differences in the prediction of single-component water isotherm (1.8 to 2x) and the breakthrough measurements, highlighting the challenges of the accurate prediction. They replied, "However, we feel that Reviewer 2's opinion that the platform is not sufficiently validated is based on her/his perception that we only assert a few percent mismatches between experimental and computational values." I respectfully submit that the difference between the measured and experimentally determined single-component water isotherm is not a few percentages by easily 1.8 to 2x. This will have a major impact on the prediction of binaries.

For my comment about the breakthrough measurements that show that there is very likely no CO₂ capacity in the presence of H₂O, the authors replied that CALF-20 will give similar results at 85% RH. This is probably true. However, as I understand, CALF-20 has weaker interactions with water at lower RH but from the isotherms MIP-22 will have a stronger interaction of water also at lower RH. Since the experimental water loadings are (much) higher than what is predicted, the actual binary loading of CO₂ (even at lower RH) is likely to be low, i.e. similar to Zeolite 13X. This means that the Water resistance coefficient (WRC) of MIP-22 (as shown in Fig S72) is likely to be closer to Zeolite 13X. The authors say that this will have a small impact on the outcomes. It is my humble opinion that this is due to the overarching simplifications used in the modelling.

My assessment of the paper remains, and I summarize. The framework provides a new way of looking at the selection of CO₂ capture materials and is useful. Other publications have considered the different aspects disparately, but they are brought to a common platform here. The framework is built on several assumptions and simplifications that must be validated adequately. If not, it will raise concerns about how much we can trust the platform to make decisions. The valiant effort to synthesize a new material is commendable; however, comparing the experimental measurements and predictions highlights the challenges in multi-scale predictions.

To put it in a single sentence: "I'm not saying that the framework is wrong. I just wish that it can be shown to be correct". I wish the authors the best in their pursuit of this challenging problem.

REVIEWER 2

Reviewer Point P 2.1 — I thank the authors for responding to my queries. In my review, I pointed out that while the synthesis of MIP-22 is appreciated, there were major differences in the prediction of single-component water isotherm (1.8 to 2x) and the breakthrough measurements, highlighting the challenges of the accurate prediction. They replied, “However, we feel that Reviewer 2’s opinion that the platform is not sufficiently validated is based on her/his perception that we only assert a few percent mismatches between experimental and computational values.” I respectfully submit that the difference between the measured and experimentally determined single-component water isotherm is not a few percentages by easily 1.8 to 2x. This will have a major impact on the prediction of binaries.

Reply: *We agree with the reviewer that the water isotherms are important and that there is a lot of uncertainty in both experimental isotherms and those predicted by molecular simulations.*

However, we differ with the reviewer on the impact of these differences in the H₂O isotherms on the estimate of the nCAC. Compared to dry flue gasses, there are two main contributions to the nCAC that are important when H₂O is considered in the flue gas stream :

- *The loss of productivity due to water in the flue gas.*

In the case of productivity, the water uptake of the sorbent at the feed conditions will impact how much of the column is inactive for CO₂ sorption. The lower the H₂O working capacity of the material, the further the water concentration front will travel into the bed. In the water impact section (SI section 9), we have modeled water penetration three times our original predicted value —this encompasses the 1.8 to 2x uncertainty mentioned by the reviewer.

- *The energy consumption required for water desorption.*

In our process design, operating variables (e.g. adsorption step times) are chosen to avoid break through of either CO₂ or H₂O. Hence, there is no accumulation of either component under operation at cyclic steady state, i.e. the amount of CO₂ or H₂O regenerated from the adsorbent in each cycle are equal to the amount of either component fed into the cycle. So if we feed 2 mol of H₂O for every 10 mol of CO₂, then we always remove 2 mol of H₂O per cycle. In this way, we ensure that our water penetration distance stays constant under cyclic steady state conditions. This imposed constraint makes the energy penalty per mole of CO₂ captured mainly dependent on the heat of adsorption of the water.

The typical differences between the heats of adsorption of different structures are far less than the corresponding differences in Henry coefficients.

Reviewer Point P 2.2 — For my comment about the breakthrough measurements that show that there is very likely no CO₂ capacity in the presence of H₂O, the authors replied that CALF-20 will give similar results at 85% RH. This is probably true. However, as I understand, CALF-20 has weaker interactions with water at lower RH but from the isotherms MIP-22 will have a stronger interaction of water also at lower RH. Since the experimental water loadings are (much) higher than what is predicted, the actual binary loading of CO₂ (even at lower RH) is likely to be low, i.e. similar to Zeolite 13X. This means that the Water resistance coefficient (WRC) of MIP-22 (as shown in Fig S72) is likely to be closer to Zeolite 13X. The authors say that this will have a small impact on the outcomes. It is my humble opinion that this is due to the overarching simplifications used in the modelling.

Reply: *The process evaluations performed in this study assume high feed relative humidity (RH). We have not made any observations regarding the materials' performance at low RH conditions. Our comment regarding the "small impact on the outcomes" refers only to materials with specific combinations of properties and not to all materials, as implied by the Reviewer.*

As already emphasized in section 9.3.1 of the SI, materials with high alpha and low WRC will be disproportionately affected by water and, accordingly, uncertainty in the water-associated parameters. However, these materials are typically performing poorly. In contrast, the top-performing materials are far less sensitive to water-related assumptions. The reason is that in our chosen process configuration, the impact of the water-related assumptions on the KPIs for top performers is far less. WRC-related impacts on productivity only apply to the proportion of a bed that is wetted during the cycle, which is below 10% for top performers. Therefore, the cases considered in this study are far more resilient to poor CO₂-H₂O selectivity than if another process configuration were to be adopted.

Reviewer Point P 2.3 — My assessment of the paper remains, and I summarize. The framework provides a new way of looking at the selection of CO₂ capture materials and is useful. Other publications have considered the different aspects disparately, but they are brought to a common platform here. The framework is built on several assumptions and simplifications that must be validated adequately. If not, it will raise concerns about how much we can trust the platform to make decisions. The valiant effort to synthesize a new material is commendable; however, comparing the experimental measurements and predictions highlights the challenges in multi-scale predictions.

To put it in a single sentence: "I'm not saying that the framework is wrong. I just wish that it can be shown to be correct". I wish the authors the best in their pursuit of this challenging problem.

Reply: *The reviewer does raise an important point. Can we guarantee that our framework gives the correct answer for the trillion possible MOFs? We have to be realistic; given the framework's complexity and the sheer number of possible materials, it would be impossible to give this guarantee for each and every material.*

However, that is not the platform's aim. By following our holistic approach early in the technology development process, we can accelerate deployment and focus experimental efforts and investments on pathways with a much higher chance of success at the commercial scale.

With the platform, we can guarantee that if we identify a large number of materials that outperform the benchmark, this is a much better starting point for the next step. This next step will involve a more careful examination of all the assumptions for top-performing materials, and at this stage, we may lose a few of them. We would also expect that we might lose materials at this step because of factors we could not screen for (stability, ease of synthesis, etc.).